



# Near-surface and columnar measurements with a Micro Pulse Lidar of atmospheric pollen in Barcelona, Spain

Michaël Sicard[1,2], Rebeca Izquierdo[3], Marta Alarcón[3], Jordina Belmonte[4,5], Adolfo Comerón[1], José Maria Baldasano[6,7]

[1]Remote Sensing Laboratory, Universitat Politècnica de Catalunya, Barcelona, Spain
[2]Ciències i Tecnologies de l'Espai - Centre de Recerca de l'Aeronàutica i de l'Espai / Institut d'Estudis Espacials de Catalunya (CTE-CRAE / IEEC), Universitat Politècnica de Catalunya, Barcelona, Spain
[3]Departament de Física, Universitat Politècnica de Catalunya (UPC), c/Urgell 187, 08036 Barcelona, Spain
[4]Departament de Biologia Animal, Biologia Vegetal i Ecologia, Universitat Autònoma de Barcelona (UAB). Edifici C, 08193 Bellaterra, Spain
[5]Institut de Ciencia i Tecnología Ambientals (ICTA), Universitat Autònoma de Barcelona (UAB). Edifici Z, 08193 Bellaterra, Spain
[6]Earth Sciences Department, Barcelona Supercomputing Center – Centro Nacional de Supercomputación, Barcelona, Spain
[7]Environmental Modeling Laboratory, Technical University of Catalonia, Barcelona, Spain

*Correspondence to*: Michaël Sicard (msicard@tsc.upc.edu)

**Abstract.** We present for the first time continuous hourly measurements of pollen near-surface concentration and lidar-derived profiles of particle backscatter coefficients and of volume and particle depolarization ratios during a 5-day pollination event observed in Barcelona, Spain, between 27 – 31 March, 2015. Daily average concentrations ranged $1082 - 2830$ pollen $m^{-3}$ $day^{-1}$. *Platanus* and *Pinus* pollen types represented together more than 80 % of the total pollen. Maximum hourly pollen concentrations of 4700 and 1200 $m^{-3}$ $h^{-1}$ were found for *Platanus* and *Pinus*, respectively. Everyday a clear diurnal cycle caused by the vertical transport of the airborne pollen was visible on the lidar-derived profiles with maxima usually reached between 12 and 15 UT. A method based on the lidar polarization capabilities was used to retrieve the contribution of the pollen to the total aerosol optical depth (AOD). On average the diurnal (9 – 17 UT) pollen AOD was 0.05 which represented 29 % of the total AOD. Maximum values of the pollen AOD and its contribution to the total AOD reached 0.12 and 78 %, respectively. The diurnal means of the volume and particle depolarization ratios in the pollen plume were 0.08 and 0.14, with hourly maxima of 0.18 and 0.33, respectively. The diurnal mean of the height of the pollen plume was found at 1.24 km with maxima varying in the range $1.47 - 1.78$ km. A correlation study is performed 1) between the depolarization ratios and the pollen near-surface concentration to evaluate the ability of the former parameter to monitor pollen release, and 2) between the depolarization ratios and surface downward solar fluxes, which cause the atmospheric turbulences responsible for the particle vertical motion, to examine the dependency of the depolarization ratios upon solar fluxes. For the volume depolarization ratio the first correlation study yields to correlation coefficients ranging $0.00 - 0.81$ and the second to correlation coefficients ranging $0.70 - 0.86$.





## 1. Introduction

Pollen is the male gametophyte of seed plants. Both gymnosperms (cone-bearing plants) and angiosperms (flowering plants) produce pollen as part of sexual reproduction. Pollen grains have characteristic walls with distinctive ornamentation that permit their identification. The production and emission of pollen are governed by interacting environmental factors, as photoperiod, temperature and water stress (Dahl et al, 2013). Pollen is primarily dispersed in the atmosphere by insects or wind. Wind-pollinated plants are called anemophilous and they produce huge amounts of pollen grains which, once airborne, are responsible of allergenic reactions when inhaled by humans (Cecchi, 2013). Fungal spores are a biological component that can be found any time of the year in the atmosphere (Lacey, 1981; Burch and Levetin, 2002). Environmental variables, such as temperature and moisture, can influence growth and reproduction in fungi which makes airborne spore concentrations to fluctuate seasonally (Grinn-Gofrón and Strzelczak, 2008; Pakpour et al., 2015). However, it has also been observed that local climate, vegetation patterns and management of landscape are governing parameters for the overall spore concentration, while the annual variations caused by weather, although not negligible, are of secondary importance (Skjøth et al., 2016). Fungi are, after pollen, the second most important producers of outdoor airborne allergens (Weikl et al., 2015). Their presence can cause human health problems (mainly allergies) and crop infections (phytopathology) (e.g. Burge and Rogers, 2000; Simon-Nobbe et al., 2008). Up to 80 % of asthmatics are sensitized to fungal allergens (Lopez and Salvaggio, 1985) and a disease pattern of severe asthma with fungal sensitization has been recently proposed (Denning et al., 2006).

Worldwide many people living in large cities suffer from allergies linked to the presence of atmospheric pollen and fungal spores. In the industrialized countries of central and northern Europe, up to 15 % of the population is sensitive to pollen allergens (WHO, 2003; Cecchi, 2013). In Europe the most common types of pollen are *Ambrosia*, *Alnus*, *Artemisia*, *Betula*, *Corylus*, Chenopodiaceae, Cupressaceae/Taxaceae, *Olea*, *Platanus*, Poaceae, *Quercus*, and *Urtica*/*Parietaria* (Skjøth et al. 2013). Although their concentration is monitored daily at ground level by aerobiological networks (Scheifinger et al., 2013, Karatzas et al., 2013), very little is known on their vertical distribution and their long/short range transport (Sofiev et al., 2013) although an increase interest has arisen very recently in the aerosol lidar community (Sassen, 2008; Noh et al., 2013a; 2013b). Sassen (2008) reported on lidar measurements in the lower atmosphere of birch pollen plumes from the boreal forest of Alaska. Noh et al. (2013a) retrieved optical properties with a polarization-sensitive lidar for *Pinus* and *Quercus* pollen in South Korea. Noh et al. (2013b) reported on the vertical distribution of the same pollen event observed with lidar and on the dependency of its diurnal variations upon the meteorological conditions (temperature, relative humidity, and wind speed).

In the Mediterranean city of Barcelona, Spain, the most abundant pollen taxa are *Quercus* (27,4 % of the total pollen), *Pinus*, *Platanus*, Cupressaceae, *Olea*, Urticaceae, Poaceae, Chenopodiaceae, *Plantago*, Moraceae, *Fraxinus*, *Castanea* and *Populus* (1 %) according to the Aerobiological Network of Catalonia (http://lap.uab.cat/aerobiologia) and based on pollen concentrations measured in the period 1994-2015. Barcelona is the Spanish city which presents the longest mean pollen season of *Platanus* pollen. Maximum daily counts occur generally during the second half of March (Díaz de la Guardia et al., 1999). *Platanus hispanica* is responsible of the most frequent pollen sensitizations (37 %) detected in Barcelona (Puiggròs et al,





2015).  *Pinus* is one of the most abundant pollen taxa in Spain (Belmonte and Roure, 1991; De Linares et al. 2014). In Barcelona *Pinus* pollination occurs in two phases, the most important one happens from March to May and the other in June-July.  Regarding airborne fungal spores, *Cladosporium* is the most abundant taxon in Barcelona, representing up to 44% of the total fungal spore spectrum. It is present all year round, and shows

the highest concentrations in April-May and November (Infante et al., 1999; see also http://lap.uab.cat/aerobiologia). This paper aims at investigating the possible correlation between pollen near-surface concentration and columnar properties measured during a 5-day pollination event in Barcelona.  The dataset is composed of continuous measurements at a temporal resolution of one hour.  The influence of the meteorological conditions and the solar radiation on the pollen dispersion is also investigated.  This contribution

relates for the first time near-surface and lidar measurements of pollen in a large European city.

The paper is organized as follows: Section 2 presents the methods used to count the pollen and spore taxa and also describes the lidar instrument used and the method employed to estimate the pollen optical properties. Pollen and spores measured in Barcelona are first analyzed with daily mean concentration values and lidar quicklooks (time-altitude contour plots) in Section 3.  The analysis of the temporal evolution of hourly

concentrations and meteorological parameters completes this Section. Section 4 is dedicated to the investigation of possible correlations between near-surface pollen concentration and the vertical distribution of a series of structural and optical properties.  Finally Section 5 enlightens us on the relationship between the vertical transport of airborne pollen and the solar radiation.

## 2.    Instrumentation and method

### 2.1.    Pollen and spores sampling instrumentation, $PM_{10}$ and meteorological data

Airborne pollen and spore data were collected by the Aerobiological Network of Catalonia at the sampling station situated on the roof of a building in the city centre of Barcelona (2.164º E, 41.394° N). Samples were obtained using volumetric suction pollen-spore trap based on the impact principle (Hirst, 1952), the standardized method in European aerobiological networks. The Hirst sampler is calibrated to handle a flow of

10 litre of air per minute, thus matching the human breathing rate. Pollen and spores are impacted on a cylindrical drum covered by a melinex film coated with a 2% silicon solution as trapping surface. The drum was changed weekly and the exposed tape was cut into seven pieces, which were mounted on separate glass slides. Pollen and spores identification was performed using a microscope equipped with a 40X/15 lens. Daily average and diurnal (hourly) pollen and spore concentrations were made following the standardized Spanish

method (Galán et al., 2007), consisting in: (1) four continuous longitudinal sweeps along the 24 h slide for daily data, which gives a subsample accounting for 12-13% of the total surface; and (2) twenty-four continuous transversal sweeps separated every 2mm along the daily-sample slide, since the tape rotates 2 mm every hour, for hourly data. Pollen and spore concentrations and diurnal variations (hourly concentrations) were expressed as the number of pollen grains and spores per cubic meters of air.




PM$_{10}$ measurements were acquired at the "Eixample" station of the Xarxa de Vigilància i Previsió de la Qualitat de l'Aire (XVPCA, the Catalonian network for monitoring and forecasting the air quality). It is located at 1.2 km to the southwest of the pollen sampling instrumentation.

Meteorological data were recorded in the "Zona Universitaria" area of Barcelona, at approximately 0.6 km
south-southeast of the lidar site.

### 2.2. The Barcelona Micro Pulse Lidar

The profiles of the particle backscatter coefficient and the particle depolarization ratio were measured with the Barcelona Micro Pulse Lidar (MPL) system, model MPL-4B. The system is located in the "Zona Universitaria" area of the city, on the roof of the Remote Sensing Lab (RSLab) building in the North Campus of the Universitat
Politècnica de Catalunya (2.112º E, 41.389º N, 115 m a.s.l.), approximately 1 km from Sierra de Collserola and 7 km from the sea. It is located at 4.4 km to the west of the pollen sampling instrumentation. The system should become very shortly part of the MPLNET (Micro Pulse Lidar Network, http://mplnet.gsfc.nasa.gov/) network. The MPL system is a compact, eye-safe lidar designed for full-time unattended operation (Spinhirne, 1993; Campbell et al., 2002; Flynn et al., 2007). It uses a pulsed solid-state laser emitting low laser pulse energy
(~6 µJ) at a high pulse rate (2500 Hz) and a co-axial "transceiver" design with a telescope shared by both transmit and receive optics. The Barcelona MPL optical layout uses an actively controlled liquid crystal retarder which makes the system capable to conduct polarization-sensitive measurements by alternating between two retardation states (Flynn et al., 2007). The signals acquired in each of these states are recorded separately and called "co-polar" and "cross-polar". In nominal operation the raw temporal and vertical resolutions are 30 s
and 15 m, respectively.

The linear volume depolarization ratio, $\delta^V$, is defined as:

$$\delta^V(z) = \frac{\beta_\perp(z)}{\beta_\parallel(z)} = \frac{P_\perp(z)}{P_\parallel(z)} \tag{1}$$

where $\beta_\perp$ and $\beta_\parallel$ denote the total (particles + molecules) perpendicular and parallel backscatter coefficient, respectively, and $P_\perp$ and $P_\parallel$ the backscatter powers. According to Gimmestad (2008) $\delta^V$ can also be expressed
as a function of a factor $d$ which has a range of 0–1 and which is related to the propensity of the scattering medium to preserve the incident polarization:

$$\delta^V(z) = \frac{d}{2-d} \tag{2}$$

In the case of a linear polarization lidar, $d = 0$ indicates that no depolarization occurs, while $d = 1$ indicates that the returned beam is completely depolarized. By adapting the notations of Flynn et al. (2007), especially in Eqs
(1.4) and (1.6), to ours one can formulate the linear volume depolarization ratio for the MPL system as:

$$\delta^V(z) = \frac{P_{cr}(z)}{P_{co}(z) + P_{cr}(z)} \tag{3}$$

where $P_{cr}$ and $P_{co}$ represent the MPL cross- and co-polar channels, respectively.



The linear particle depolarization ratio, $\delta^p$, can then be determined by (Freudenthaler et al., 2009):

$$\delta^p(z) = \frac{\beta^p_\perp(z)}{\beta^p_\parallel(z)} = \frac{\left[1+\delta^m\right]\delta^V(z)R(z) - \left[1+\delta^V(z)\right]\delta^m}{\left[1+\delta^m\right]R(z) - \left[1+\delta^V(z)\right]} \qquad (4)$$

where $\beta^p_\perp$ and $\beta^p_\parallel$ are the particle perpendicular and parallel backscatter coefficients, respectively, $\delta^m$ is the molecular depolarization ratio and $R$ is the backscatter ratio which is defined as:

$$R(z) = \frac{\beta^m(z) + \beta^p(z)}{\beta^m(z)} \qquad (5)$$

where $\beta^m$ and $\beta^p$ denote the molecular and particle backscatter coefficient, respectively, of the total (perpendicular + parallel) returned signal. According to the MPL optical requirements in the receiving system the spectral filtering is performed by filters with a spectral band $\leq 0.2$ nm. This number produces a temperature-independent molecular depolarization ratio of $\delta^m = 0.00363$ according to Behrendt and Nakamura (2002).

The particle backscatter coefficient, $\beta^p$, was retrieved with the two-component elastic algorithm (also known as the Klett-Fernald-Sasano method; Fernald, 1984; Sasano and Nakane, 1984; Klett, 1985) with a constant lidar ratio of 50 sr and applied to the total lidar signal, P, reconstructed from the MPL lidar signals as (Flynn et al., 2007):

$$P(z) = P_{co}(z) + 2P_{cr}(z) \qquad (6)$$

The value of 50 sr is motivated by two previous studies. First, it falls in the range of the mean columnar lidar ratios, $46 - 69$ sr, found in Barcelona during the period from February to April and calculated over a period of 3 years (Sicard et al., 2011). In that work the columnar lidar ratio was retrieved with the two-component elastic lidar inversion algorithm constrained with the aerosol optical depth from a sun-photometer (Landulfo et al., 2003; Reba et al., 2010). Second, Noh et al. (2013b) used the same method and found a mean columnar lidar ratio of $50 \pm 6$ sr during a 6-day pollination event (mostly dominated by *Pinus* and *Quercus* pollen) in South Korea. At the peak of the event the pollen aerosol optical depth (AOD) represented up to 35 % of the total AOD.

### 2.3. Determination of pollen optical and structural properties

Pollen has formerly been distinguished from other particle types thanks to its depolarization capabilities (Sassen, 2008; Noh et al., 2013a; 2013b). Although many types of pollen have regular shapes (circular, spherical, elliptical, ovoid, etc.), they cannot be considered spherical from the point of view of light scattering because they do not generate Mie patterns expected from a sphere of equivalent size. The reason lies in surface imperfections of pollen grains and inhomogeneous refractive indices inside the grains

When the atmospheric particle load can be assumed as the external mixing of one type of depolarizing particles (the pollen, here) with another type of non-depolarizing particles, the method suggested by Shimizu et al. (2004) allows to separate the contribution ratio of both types of particles. In our case, the pollen (depolarizing particles) contribution ratio, $CR_{pol}$, can be expressed as:



$$\mathrm{CR}_{pol}(z) = \frac{\left[\delta^p(z) - \delta_{no-pol}\right]\left[1 + \delta_{pol}\right]}{\left[\delta_{pol} - \delta_{no-pol}\right]\left[1 + \delta^p(z)\right]} \qquad (7)$$

where $\delta_{no-pol}$ and $\delta_{pol}$ are the particle depolarization ratios of all particle types except pollen (non-depolarizing) and the pollen (depolarizing), respectively. One can check easily that if no pollen is present $\delta^p = \delta_{no-pol}$ and Eq. (7) leads to $CR_{pol} = 0$, and that if only pollen is present $\delta^p = \delta_{pol}$ and Eq. (7) leads

then to $CR_{pol} = 1$. The pollen backscatter coefficient, $\beta_{pol}$, is simply calculated as:

$$\beta_{pol}(z) = CR_{pol}(z)\beta^p(z) \qquad (8)$$

The contribution ratio is sensitive to the selection of $\delta_{no-pol}$ and $\delta_{pol}$ which are determined either empirically or taken from references. To fix the value of $\delta_{no-pol}$ we searched for a clear-sky day prior to the pollination event without long-range transport aerosols. Such conditions were fulfilled on 15 March around 12 UT. On

that day a well-mixed atmospheric boundary layer (ABL) developed. At 12 UT the ABL height was ~1.2 km and the AOD 0.18. The particle depolarization ratio was constant ~0.03 in the whole ABL. We have taken $\delta_{no-pol} = 0.03$. In the literature very few information on measurements of pollen depolarization ratios is available. The choice of $\delta_{pol}$ is deferred to Section 4, after the analysis of the individual profiles of $\delta^p$, in order to have as much information as possible.

Finally we also calculated the vertical height, $h_{pol}$, up to which the pollen plume extends. As it is shown in Section 4, the pollen plume is characterized during the whole pollination event by a near-constant or slightly decreasing profile of $\beta_{pol}$. From this aspect the structure of the pollen plume is much simpler than the ABL structure usually found in Barcelona (Sicard et al., 2006). This allows us to use a simple threshold method such as the one used to estimate the ABL height by Melfi et al. (1985) and Boers et al. (1988). After several tests

we empirically set a threshold of 0.055 Mm$^{-1}$ sr$^{-1}$ and defined $h_{pol}$ as the first height at which

$$\beta_{pol}(z) < 0.055\, Mm^{-1} sr^{-1}.$$

### 3. Temporal variation of pollen and spore taxa near-surface and columnar properties

In the second half of March 2015 a strong anticyclone positioned in the Atlantic Ocean west of the Portuguese coast generated southeasterly winds in the northeastern part of the Iberian Peninsula. In Barcelona, the synoptic

conditions resulted in marked, off-shore winds in altitude yielding to relatively clear skies and preventing long-range transport of highly depolarizing aerosols like mineral dust over Barcelona.

Ninety three pollen types and forty fungal spore types are counted routinely on a daily basis at the Aerobiological Network of Catalonia. The daily variation of the concentration of the four most abundant pollen (*Platanus, Pinus* and Cupressaceae) and spore (*Cladosporium*) taxa and the total (pollen + spore) is represented





in Figure 1a for a period surrounding the peak of the pollination event under study. Figure 1b shows the fraction of each one of the four most abundant taxa to the total (pollen + spore). During the pollination event, 26 – 31 March, the total concentration varies between 1082 and 2830 pollen and fungal spore per cubic meter. Three days before (23 March) and after (3 April) the event, values of 275 and 368 $m^{-3}$ $day^{-1}$ are registered, respectively.

5 The most abundant taxon is *Platanus* which represents between 48 – 71 % of the total concentration during the pollination event. This range of values is higher than the annual fraction of *Platanus* to total pollen, 46.3 %, estimated by Gabarra et al. (2002) over the period 1994 – 2000 in the city of Barcelona. The *Platanus* daily concentration reaches a maximum of 1703 $m^{-3}$ $day^{-1}$ on 31 March. This value is in the lower part of the range of daily maxima (1543 – 2567 $m^{-3}$ $day^{-1}$) observed per year over the period 1994 – 2000 by Gabarra et al. (2002).

10 *Pinus* is the second most abundant taxon which represents between 18 and 30 % of the total concentration during the pollination event and reaches a maximum of 803 $m^{-3}$ $day^{-1}$ on 30 March. During the whole event *Platanus* and *Pinus* pollen types represent 80 % or more of the total concentration. *Pinus* is the taxon that presents the highest relative increase since its fraction passes from values lower than 10 % before the event to up to values ranging in 18 -30 % during the event. The third most abundant taxon is *Cladosporium* spore which

15 represent between 6 and 11 % of the total concentration during the pollination event and reaches a maximum of 224 $m^{-3}$ $day^{-1}$ on 31 March. This value is of the order of magnitude of the daily means observed during the month of March (~ 200 $m^{-3}$ $day^{-1}$) by Infante et al. (1999) over a 6-year period in the city of Barcelona. Finally the fourth most abundant taxon is Cupressaceae which does not count for more than 5.4 % (on 27 March) of the total concentration. With a maximum peak of 74.9 $m^{-3}$ $day^{-1}$ (on 28 March), this event is of rather low

20 intensity for Cupressaceae as it falls at the end of the pollen season for that taxon according to Belmonte et al. (1999).

The temporal evolution of the profiles of the particle backscatter coefficient and the volume depolarization ratio during the pollination event is shown in Figure 2. Aerosols are present everyday up to 2.5 – 3 km. However most of the aerosol load is found below approximately 1.5 km. Near the ground (< 0.5 km) high values of $\beta^p$

25 (4 $Mm^{-1}$ $sr^{-1}$) are found on almost all days. Between 0.5 and 1.5 km the green color code indicates values of $\beta^p$ not higher than 2 – 2.5 $Mm^{-1}$ $sr^{-1}$ (except on 26 March on which day clouds are present below 2 km before 08 UT). In general two regimes are observed everyday: an increase in amplitude and height starting around 10 UT which persists until the night, and a less pronounced nighttime regime starting usually after midnight. On 31 March one sees a layer appearing after 11 UT with very large values of $\beta^p$ (> 5 $Mm^{-1}$ $sr^{-1}$) and confined in the

30 first 0.5 km of the ABL. This increase of $\beta^p$ in the bottom part of the ABL has no impact on the volume depolarization ratio vertical distribution (Figure 2b) which suggests that it is due to non-depolarizing particles. The green color code volume depolarization ratio shown in Figure 2b indicates values of $\delta^V$ near 0.02 – 0.03. It is the usual value of $\delta^V$ for background, local aerosols near the surface in Barcelona. Everyday around 08 UT a plume with $\delta^V$ > 0.08 (yellowish) appears, raises up to 1.0 – 1.7 km in a few hours and starts decreasing

35 before 16 UT at a lesser rate than it raised. This diurnal pattern of $\delta^V$ is observed on each single day of the pollination event. On the first four days values of $\delta^V$ larger than 0.08 are no longer detected after 18 – 20 UT. Toward the end of the event on 30 and 31 March when the pollen concentrations were the highest values of $\delta^V$ > 0.08 are still detected until 21-24 UT. The highest values of $\delta^V$ are detected on 30 March and are of the order of 0.22. This maximum value is higher than the peak value of 0.15 observed by Noh et al. (2013b) for *Pinus*



and *Quercus* pollen in South Korea and lower than $\delta^V = 0.30$ measured by Sassen (2008) for birch pollen plumes from the boreal forest of Alaska.

Because of the presence of clouds near the ABL top in the first part of 26 March (Figure 2b), this day is discarded in the rest of the paper and from now on we will focus only on the period 27 – 31 March. To gain an insight into the temporal variations of the pollen and spore taxa concentration we performed for the period 27 – 31 March a counting on an hourly basis. The method used is described in Section 2.1. Although all pollen and spore taxa were counted, in the following we will only show the results of the total pollen (spore is no longer taken into account) and of the two most abundant pollen types: *Platanus* and *Pinus*. The two main reasons for that choice are that 1), as found earlier, *Platanus* and *Pinus* pollen represent more than 80 % of the total (pollen + spore) taxa and 2) the ratio of total spore to total pollen is less than 13 % during the period 27 – 31 March.

Many works have investigated the influence of the meteorological conditions, such as relative humidity, temperature, wind speed, the number of sunshine hours and rainfall, on the release and transport of pollen in the atmosphere (Raynor et al., 1973, Mandrioli et al., 1984; Hart et al., 1994; Alba et al., 2000; Jato et al., 2000; Bartková- Ščevková, 2003; Vázquez et al., 2003; Latorre and Caccavari, 2009, among others). On the one hand, relative humidity and temperature greatly affect the release of pollen in the atmosphere by influencing the extent to which individual pollen grains dehydrate. For example a low relative humidity associated with a high temperature will tend to increase the number of airborne pollen grains by decreasing their specific gravity. The relation of pollen with water comes from its hydrophilic properties and from the fact that it is prone to harmomegathic movement (accommodation of volume change when it absorbs water; Wodehouse, 1935). The duration of the sunshine has also proved to have an influence on the pollen release (Alba et al., 2000). On the other hand, wind speed plays a major role in the transport and dispersion of airborne pollen: high daytime wind speed may facilitate the dispersion of airborne pollen in the atmosphere (Latorre and Caccavari, 2009; and references therein). The effect of rainfall is to reduce the number of airborne pollen grains by washing out the atmosphere. During the pollination event presented in this work, no rain was detected.

In Figure 3 we present the hourly temporal variations of 1) *Platanus*, *Pinus* and total pollen concentration during the period 27 – 31 March, together with 2) relative humidity (RH), 3) temperature (T) and 4) wind speed. In Figure 3b we also indicated the time of the maximum pollen concentration and pollen AOD on each day. The pollen AOD, $AOD_{pol}$, was obtained by integrating the profile of $\beta_{pol}$ from the ground up to $h_{pol}$ and multiplying the result by the same lidar ratio used in the lidar inversion, 50 sr (see Section 2.2). With the exception of 31 March, the pollen number concentration at ground level follows a clear diurnal cycle (Figure 3a). On 31 March, no clear difference is observed between day and night. *Platanus* and *Pinus* concentration reaches maximum peaks of ~4700 m$^{-3}$ h$^{-1}$ on 31 March and 1200 m$^{-3}$ h$^{-1}$ on 30 March, respectively. Maximum peaks of the total pollen concentration higher than 5000 and 6000 m$^{-3}$ h$^{-1}$ are reached on 30 and 31 March, respectively. They are associated with absolute peaks of *Platanus* and relative peaks of *Pinus*. Interestingly a release cycle is visible each day: the diurnal variation is marked with several relative peaks along the day that are usually distant in time by 2 to 4 hours. *Platanus* and *Pinus* peaks are not necessarily correlated. The fact that *Platanus* variations are shaper than *Pinus* ones may be explained by the size difference between both pollen





types: while *Platanus* longest diameter (on the polar axis) varies between 21 and 28 µm, it varies between 60 and 74 µm for *Pinus* (https://www.polleninfo.org/AT/en/allergy-infos/aerobiologics/pollen-atlas.html?letter=P). Pollen size is known to be a factor affecting pollen release but also their settlement to the ground (McCartney, 1994).

The relative humidity and temperature hourly evolution shows a clear diurnal cycle (Figure 3b): a relative humidity decrease associated with a temperature increase is observed during daytime while the opposite occurs during nighttime. Daytime RH (T) values are found in the range of 40 – 60 % (17 – 25 ºC) while nighttime values are found in the range of 65 – 90 % (12 – 18 ºC). While no marked trend is observed on the relative humidity along the pollination event, a temperature day-to-day increase is observed, the daily mean temperature

passing from 15.2 – 17.5 – 16.6 – 18.5 to 17.9 ºC between 27 and 31 March. The correlation coefficient between the daily mean temperature and the daily total pollen concentration is 0.95, indicating a strong dependence of pollen release upon temperature. The correlation coefficient between the daily mean relative humidity and the daily total pollen concentration, -0.18, is negative but much lower (in absolute value) than the one for temperature. Except on 28 and 31 March (when wind speeds higher than 6 m s$^{-1}$ are detected in the first half of

the day), the daytime wind speed usually oscillated between 2 and 3 m s$^{-1}$ (with gusts at ~4.5 m s$^{-1}$) which corresponds to a light breeze. From 27 to 31 March, the daily wind speed varies from 1.6 – 2.5 – 1.5 – 3.7 to 2.4 m s$^{-1}$, similarly to the daily mean temperature. The correlation coefficient between daily wind speed and total pollen concentration is 0.82, indicating a strong dependence of pollen release also upon wind speed.

Each day the time of the maximum peak of the total pollen concentration (red vertical lines, Figure 3b) occurs

between 02 and 11 UT, while that of AOD$_{pol}$ (grey vertical lines, Figure 3b) occurs more regularly between 12 and 15 UT. As expected, every day the total pollen concentration peak precedes the AOD$_{pol}$ peak. Logically a strong release of pollen at the ground level at a given time is chronologically followed by a peak of the amount of pollen in the atmosphere, here parameterized by the pollen AOD, when the conditions for the dispersion of pollen in the atmosphere are gathered. On the one hand and surprisingly the total pollen concentration peaks

are not systematically associated with minima of RH, maxima of T and/or of the wind speed, while, on the other hand, the pollen AOD peaks are systematically associated with minima of RH and maxima of T. The pollen AOD peaks do not present a systematic dependence upon wind speed, a result in agreement with the findings of Noh et al. (2013a) who showed a broad variation of the pollen AOD for wind speeds lower than 3 m s$^{-1}$.

## 4. Pollen near surface vs. columnar properties: day-by-day analysis

The daily temporal variation of some lidar-derived range-resolved and columnar parameters are investigated and further compared to pollen concentrations in order to find possible correlations. Figure 4 shows the diurnal (9 – 17 UT) profiles of the particle and pollen backscatter coefficients and of the volume and particle linear depolarization ratios for the 5 days of the event. The top height of the pollen plume, $h_{pol}$, is also indicated in the plots by horizontal grey lines. The profiles of $\delta^{V}$ are characterized by a near-constant or slightly decreasing

slope with increasing height which reaches zero generally sharply at $h_{pol}$. The profiles of $\delta^{p}$, which unlike



$\delta^v$ show only the particle depolarization effect, have a general tendency to decrease with increasing height, reflecting the gradual diminution of the number of pollen as height increases. It is also frequent to find $\delta^p > 0.2$, especially at the beginning of the day, in the lowermost part of the ABL (below 0.3 km) where most of the pollen grains concentrate.

Maxima of $\delta^v$ of the order of 0.22 are reached on 30 March at 12 UT below 0.75 km. On that particular profile, some values of $\delta^v$ are associated with comparatively low values of $\beta^p$ ($< 2$ Mm$^{-1}$ sr$^{-1}$), and therefore with low values of the backscatter coefficient $R$ which altogether contribute to increase $\delta^p$ according to Eq. (4). This produced values of $\delta^p$ in the range of 0.40 – 0.43 at the height of 0.5 km. Other maxima of $\delta^p$ of 0.35 are observed on 30 March at 11 UT, and of 0.31 on 29 and 30 March at 12 and 13 UT, respectively, all at the same

height of ~0.5 km. With this in mind we now come back to the choice of $\delta_{pol}$ needed for applying Shimizu's method and left in stand-by in Section 2.3. Very few information is available on that subject (in chronological order):

- Sassen (2008) found maxima of $\delta^v$ of 0.30 for birch pollen plumes from the boreal forest of Alaska that he described as "unusually high for aerosols, and […] comparable to irregularly-shaped desert

dust particles raised by dust storms". This finding implies that $\delta^p > 0.30$ in the pollen cloud he observed.

- In two consecutive papers Cao et al. (2010) and Roy et al. (2011) measured the linear particle depolarization at four wavelengths of several types of pollen in an aerosol chamber with a polarization-sensitive lidar. For *Pinus* (*Platanus* was not tested) they found a mean $\delta_{pol}$ of 0.41 and 0.42,

respectively.

- Noh et al. (2013a) used Shimizu's method with a pollen depolarization ratio equal to that of pure mineral dust, $\delta_{pol} = 0.34$, without further justification.

- Noh et al. (2013b) found a maximum value of $\delta^p$ of 0.23 in a cloud of *Pinus* and *Quercus* pollen mixed with local aerosols (urban haze) in South Korea. Although they used a definition of the particle

depolarization ratio different from ours, we corrected the maximum value found in their paper with their Eq. (2) in order to make $\delta^p = 0.23$ compatible with our definition. Given their estimation of urban haze depolarization ratio, 0.03, the value of $\delta^p = 0.23$ implies $\delta_{pol} \geq 0.23$.

- In the present study we found maxima of $\delta^p$ of 0.31, 0.35 and 0.43 in a cloud of *Platanus* and *Pinus* pollen mixed with local urban aerosols. Given our estimation of the local, urban aerosol depolarization

ratio, 0.03 (see Section 2.3), the former rationale implies that $\delta_{pol}$ might be greater than 0.43, error bars apart.





It is worth noting that the maximum value of $\delta^p$ of 0.43 observed on 30 March at 12 UT coincides in time with the lowest relative humidity and the highest temperature observed during the whole pollination event (see Figure 3b). We have checked that the relative humidity and temperature profiles (not shown) measured daily by radiosoundings launched close to the lidar site were, respectively, the lowest (RH < 40 % up to 1 km) and the highest (T > 15ºC up to 1 km) on 30 March at 12 UT. It also follows a strong peak of pollen release at ground level at 09 UT (Figure 3a) that occurred at the end of an 8-hour period of strong winds of 6 to 10 m s$^{-1}$ that might have partially cleaned the atmosphere. All in all it is reasonable to think that the aerosol load on 30 March at 12 UT in the first kilometre may be composed of quasi-pure pollen. Another point to take into account is the error bar associated to the calculation of the MPL particle linear depolarization ratio. To have an estimation of this error we calculated the standard deviation of the 120 profiles that composed the 1-hour averaged profile shown in Figure 4. For the measurement of 30 March at 12 UT we find a standard deviation of 0.02 at 0.5 km and of 0.08 at 1.0 km. Given all the above, we fixed a pollen depolarization ratio of $\delta_{pol} = 0.40$.

The profile of the pollen backscatter coefficient retrieved with Shimizu's method (see Section 2.3) and $\left(\delta_{pol} = 0.40, \delta_{no-pol} = 0.03\right)$ has, like the profiles of $\delta^p$, a general tendency to decrease with increasing height, reflecting the gradual diminution of the number of pollen as height increases. In many profiles the backscatter coefficient is higher in the first 0.5 km due to the presence of a number of pollen grains larger near the ground than in altitude.

Figure 5 shows the daily cycle of the days of the pollination event in terms of pollen concentration (total, *Platanus* and *Pinus*), PM$_{10}$, and a series of column-integrated lidar-derived parameters (AOD, AOD$_{pol}$, AOD$_{pol}$ / AOD, $\overline{\delta^V}$, $\overline{\delta^p}$ and $h_{pol}$). The AOD was obtained by integrating the profile of $\beta^p$ in the whole column and multiplying the result by the lidar ratio of 50 sr used in the lidar inversion. In order to be representative of the pollen plume $\overline{\delta^V}$ ($\overline{\delta^p}$) was obtained by integrating the profile of $\delta^V$ ($\delta^p$) from the ground up to $h_{pol}$, thereby limiting the integration to the pollen plume. Table 1 gives the daily (0 – 24 UT) and diurnal (09 – 17 UT) means of all the aforementioned parameters. It is worth commenting several aspects of the daily pollen variation first. The daily and diurnal correlation coefficients, the *r*-values, of PM$_{10}$, AOD, AOD$_{pol}$, AOD$_{pol}$ / AOD, $\overline{\delta^V}$, $\overline{\delta^p}$ and $h_{pol}$ with the total pollen, *Platanus* and *Pinus* concentration is given in Table 2, 3 and 4, respectively.

Qualitatively, two classes of days can be distinguished: the days with no (or low) nocturnal pollen near-surface activity (27 and 29 March) and the days with nocturnal pollen activity (28, 30 and 31 March). One sees that when nocturnal pollen activity is observed, high pollen concentrations are reached during the night (> 4000 m$^{-3}$ h$^{-1}$) and that on two days (28 and 31 March) the nocturnal peaks are higher than the diurnal ones. There seems to be clear indications that the nocturnal pollen activity is linked to the surface wind speed (Figure 3b): 1) the three nights with nocturnal pollen activity correspond to the days with the highest wind speed daily means (see Section 3) and 2) on two nights (28 and 30 March) the nocturnal pollen activity coincides with the two periods



of highest wind speeds (> 6 m s$^{-1}$). While the total pollen concentration diurnal mean is always higher than the daily mean (Table 1), the nocturnal pollen activity is able to reverse punctually this relationship for the *Platanus* and *Pinus* concentrations. A feature common to all days is the decrease of the pollen activity between 17 UT and midnight. The hourly temporal evolution of the PM$_{10}$ is quite constant from one day to another during all

five days of the pollination event. No significant differences are noticeable. We note that the PM$_{10}$ values are well below the hourly mean values averaged for the month of March by Querol et al. (2001) in Barcelona which oscillate roughly between 30 and 70 µg m$^{-3}$ h$^{-1}$. The various correlation coefficients calculated for PM$_{10}$ do not yield to conclusive results: over the whole period (27 – 31 March) the dependence of PM$_{10}$ and pollen concentration is rather negative (positive) on a daily (diurnal) basis, but this relationship is not systematic on a

day-by-day analysis. It is important to note that the overall positive diurnal *r*-values are the result of a circumstantial dependence: the pollen number morning increase and the everyday traffic PM$_{10}$ peak occur at the same period of the day but are not linked one to another. Finally we also want to point out that PM$_{10}$ samplers have a cut-off aerodynamic diameter at 10 µm, which is significantly smaller than the diameter of the most abundant pollen grains observed (*Platanus* and *Pinus*), thus implying that no marked correlation should

be expected between PM$_{10}$ and pollen concentration.

We now move on to the analysis of the lidar-derived columnar parameters and their possible correlations with the near-surface pollen concentration. The daily mean AOD which varies in the range 0.14 – 0.24 (Table 1) is in the range of average values for this period of the year as reported by Sicard et al. (2011). AOD$_{pol}$ shows a clear diurnal cycle with maxima between 12 and 15 UT. The highest value of AOD$_{pol}$, 0.12, is reached on 30

March at 13 UT. The daily cycle and the values found here for AOD$_{pol}$ are similar to those of Noh et al. (2013a) in a cloud of *Pinus* and *Quercus* pollen observed in South Korea. The comparison of the plots of AOD and AOD$_{pol}$ in Figure 5 clearly indicates that the everyday AOD increase usually starting after 09 UT is due to the airborne pollen. The contribution of AOD$_{pol}$ to the AOD passes every day from values below 20 % before 09 UT to maxima ranging from 28 to 78 % reached between 11 and 15 UT. On 29 and 30 March the maxima

reach 61 and 78 %, respectively, while on the three other days AOD$_{pol}$ / AOD stays below 40 %. The former maxima are quite high and suggest a strong dispersion of the pollen grains in the atmosphere. Obviously the daily and diurnal means ranging, respectively, in 11 – 23 and 20 – 40 % (Table 1) are much lower than the hourly peaks. It is also interesting to note that on the last four days of the event the nighttime values of AOD$_{pol}$, although low (< 0.03), are not negligible. The daily evolutions of $\overline{\delta^V}$ and $\overline{\delta^p}$ are quantitatively very similar.

Like for AOD$_{pol}$ a clear diurnal cycle is visible everyday with maxima of 0.18 (0.33) for $\overline{\delta^V}$ ($\overline{\delta^p}$) reached on 30 March at 12 UT. The diurnal mean of $\overline{\delta^V}$ ($\overline{\delta^p}$) averaged over the whole pollination event is 0.08 (0.14). Here again the nighttime values of $\overline{\delta^V}$ ($\overline{\delta^p}$) are found non-negligible on the last four days of the event. Non-negligible nighttime values of $\delta^V$ were already observed in the time-range plots of Figure 2. Interestingly those four days correspond to the three days previously classified as days with nocturnal pollen near-surface

activity, plus 29 March that was classified as a day without nocturnal pollen activity. On 29 March the non-negligible nighttime values of AOD$_{pol}$, $\overline{\delta^V}$ and $\overline{\delta^p}$ coincide in time with a developed pollen plume (last plot





of Figure 5). Indeed $h_{pol}$ reaches its maximum nighttime peak, at 0.81 km, on 29 March at 04 UT. This observation suggests that near-surface pollen release and pollen plume dispersion in the atmosphere are not necessarily timely correlated. The daily evolutions of $h_{pol}$ are quite similar from one day to another. Heights below 0.81 km are found before 08 UT and maxima ranging in 1.47 – 1.78 km are usually reached between 14

– 16 UT. Over the whole event the diurnal mean pollen height is 1.24 km. The diurnal increase of $h_{pol}$ is smoother than that of $AOD_{pol}$, $\overline{\delta^V}$ and $\overline{\delta^p}$. As an example, let's take $\delta^V$ for the rationale. $\overline{\delta^V}$ reaches a maximum almost systematically everyday at 12 UT. While the pollen plume keeps rising vertically between 12 and 14 – 16 UT, $\overline{\delta^V}$ decreases, evidencing a dilution effect of the atmospheric pollen: the rate of the pollen vertical distribution increase is higher than that of the release of new pollen grains, if any, in the atmosphere.

Finally the sharp decrease of $h_{pol}$ on 31 March at 18 UT is an indication of the sudden removal of the pollen from the ABL. The fact that it is associated to a strong increase of the AOD but to any increase of the near-surface $PM_{10}$ level suggests that the new non-depolarizing aerosol plume seen in Figure 2 with high values of $\beta^p$ (> 5 Mm$^{-1}$ sr$^{-1}$) and confined in the first 0.5 km is not from local origin.

We have calculated the correlation coefficients between the near-surface pollen concentration (total, *Platanus*

and *Pinus*) and the columnar properties discussed above in this Section (Table 2, 3 and 4). For each parameter the highest, positive *r*-values have been colored in red. The highest daily (0 – 24 UT) *r*-values of the total pollen are on 27 March. Excepting the $PM_{10}$ parameter the same occurs for *Pinus*, whereas for *Platanus* most of the highest *r*-values are found on 29 March. Although *Pinus* is found in lesser amounts than *Platanus*, it seems to influence strongly the total pollen columnar properties. The highest diurnal (9 – 17 UT) values are

found in majority on 29 March for the total pollen and *Platanus* and on 31 March for *Pinus*. This time, during the strong diurnal pollen release, the total pollen columnar properties seem to be driven by the ones of *Platanus*. For each day, the highest, positive *r*-values have been stressed in bold font. Independently of the pollen type $\overline{\delta^p}$ appears clearly to be the parameter with the highest daily *r*-values. In particular for the total pollen and *Platanus* the daily *r*-values range between 0.39 and 0.81, leaving apart 30 March. Concerning the diurnal *r*-

values, the results are not so clear: $\overline{\delta^p}$ still appears as the parameter with the highest *r*-values for *Platanus*, but no parameter clearly stands out for the total pollen and *Pinus*.

In the concern to find a possible proxy of the pollen (be it total, *Platanus* or *Pinus*) near-surface concentration with some columnar parameter easily measurable by remote sensing instrument, we examine the correlation between pollen concentration and AOD. The AOD is a columnar parameter which is relatively easily

measurable, e.g. with a sun-photometer, and which can be retrieved with a relatively high accuracy. For example in the AERONET (Aerosol Robotic Network; http://aeronet.gsfc.nasa.gov/) worldwide network of sun/sky-photometers, the accuracy is ± 0.02 (Eck et al., 1999). Table 2, 3 and 4 show that the *r*-values of the AOD parameter, although in majority negative, can frequently change sign and reach values close to zero. The AOD is therefore not an appropriate proxy for the pollen near-surface concentration. It is interesting to note,




en passant, that $AOD_{pol}$ $r$-values are not much better and that $AOD_{pol}$ would not be a good proxy either. This result emphasizes again that near-surface pollen release and columnar pollen dispersion are not timely correlated. The correlation between pollen concentration and $\delta^V$ is also investigated because $\delta^V$ is a lidar product relatively simple to retrieve since it does not require post-acquisition processing. In the case of the

MPL system used in this study, $\delta^V$ is obtained by a very simple operation with the two collected powers (see Eq. (3)). Although $\overline{\delta^V}$ $r$-values are usually a little lower than $\overline{\delta^p}$ values, the values indicate globally a rather positive correlation between $\overline{\delta^V}$ and the pollen concentration. To investigate further the pros and contras of using $\overline{\delta^V}$ instead of $\overline{\delta^p}$, the retrieval of which is much less straight-forward than the retrieval of $\overline{\delta^V}$, we present in Figure 6 the scatter plots of the daily $(0 - 24$ UT) values of pollen concentration vs. $\overline{\delta^V}$ and $\overline{\delta^p}$. In

this figure the positive slope of the red linear regression line is a clear indicator of the positive correlation between $\overline{\delta^V}$ / $\overline{\delta^p}$ and the pollen concentration. Over the whole event the $\overline{\delta^p}$ ($\overline{\delta^V}$; $\overline{\delta^V} - \overline{\delta^p}$) $r$-values are 0.41 (0.34; -0.07) for total pollen, 0.36 (0.28; -0.08) for *Platanus* and 0.46 (0.42; -0.04) for *Pinus*. Thus overall the $\overline{\delta^V}$ $r$-values are between -0.08 and -0.04 smaller than the $\overline{\delta^p}$ ones. The highest $\overline{\delta^V}$ and $\overline{\delta^p}$ $r$-values are reached for *Pinus* ($0.09 < \overline{\delta^V}$ $r-values < 0.70$ and $0.25 < \overline{\delta^p}$ $r-values < 0.68$), while the lowest $r$-values

are reached for *Platanus* ($0.02 < \overline{\delta^V}$ $r-values < 0.68$ and $0.12 < \overline{\delta^p}$ $r-values < 0.70$). One also sees that the days previously classified as days with nocturnal pollen near-surface activity (28, 30 and 31 March) have the lowest $r$-values. This is mostly due to the points with high concentration and low depolarization ratios visible above the linear regression line, reflecting the nighttime situation of high pollen release without vertical dispersion. For the days classified as days without nocturnal pollen activity the difference between $\overline{\delta^V}$ and

$\overline{\delta^p}$ $r$-values are in general smaller than 0.04, so that it becomes nearly equivalent to use $\overline{\delta^V}$ instead of $\overline{\delta^p}$. For these days the highest $\overline{\delta^V}$ $r$-values are reached for the total pollen. Let's note that the total pollen release and dispersion is especially well correlated on 27 March ($\overline{\delta^V}$ and $\overline{\delta^p}$ $r$-values are equal to 0.81). All in all these results suggest that:

- In all conditions:

25          o   Differences between $\overline{\delta^V}$ and $\overline{\delta^p}$ $r$-values range between 0.08 and 0.04.

         o   $\overline{\delta^V}$ seems to be a proxy better for *Pinus* and the total pollen concentration than for *Platanus* concentration.

- Without nocturnal pollen near-surface activity:

         o   $\overline{\delta^V}$ and $\overline{\delta^p}$ are nearly equivalent ($r$-values differences smaller than 0.04).





      o   $\overline{\delta^v}$ seems to be an appropriate proxy for the total pollen concentration.

It is important to recall that these conclusions have to be taken in a general sense, and that, depending on the meteorological conditions, there may be cases for which these statements do not apply. In the following Section we seek the possible reasons which could explain the aforementioned correlations between pollen near-surface

concentration and columnar depolarization ratios.

## 5. Influence of the solar radiation on the pollen vertical transport in the atmosphere

For the pollen to be dispersed in the atmosphere, vertical transport is needed. According to Mandrioli et al. (1984) the main mechanism driving the vertical movement of atmospheric particles is the atmospheric turbulence. In the ABL atmospheric turbulences result from the vertical movement of air masses due to the

heating and cooling of the ground by the sun and to the flow of air (wind) over the ground. At the end of Section 3 we showed that the pollen AOD did not present a systematic dependence upon wind speed, because of the low wind speeds (usually lower than 3 m s$^{-1}$) detected during the pollination event. Thus we examine the possible influence of solar radiations on the vertical transport of pollen thanks to MPL co-located pyranometer solar flux measurements performed at the Barcelona SolRad-Net (Solar Radiation Network, http://solrad-

net.gsfc.nasa.gov/) site. The pyranometer is a Kipp and Zonen CMP21 sensor that provides every two minutes a measurement of the total solar flux in the range 0.3 – 2.8 μm. We used SolRad-Net level 1.5 data which have been cleared as free of any operational problems. The solar flux (also called solar irradiance) measured at ground level is the power per unit area produced by the sun in the form of electromagnetic radiation measured at the Earth's surface after atmospheric absorption and scattering.

Figure 7 shows the solar fluxes as a function of time for all five days of the pollination event. On 29 and 30 March clouds alter significantly the diurnal pattern of the solar radiation received at ground level. We have checked on the profiles of the MPL the presence of clouds and their altitude. On 29 March medium- and high-level clouds were present along the day in the range of 5 to 12 km, while on 30 March high-altitude clouds between 9 and 13 km were present along the day. The most part of the days of 27 and 28 March were free of

clouds. On 27 March high-level clouds in the range 9 – 12 km were present until 09:30 UT, while on 28 March clouds in the range 8 – 10 km were present until 10 UT and again after 17 UT. On 31 March the sky was totally free of clouds with the exception of clouds forming in the ABL from 17 UT onwards. The possible influence of the solar radiations on the vertical transport of pollen is examined with the clear-sky days of 27, 28 and 31 March. In the first row of Figure 8 we represent all together the solar fluxes, $\overline{\delta^v}$ and $\overline{\delta^p}$ as a function of time.

Everyday a diurnal pattern is clearly visible on all curves with an increase in the morning and a decrease in the afternoon. On 27 March the temporal evolution of $\overline{\delta^v}$ and $\overline{\delta^p}$ seems to follow especially well the pattern of the solar flux. In all three cases a time delay is observed between the diurnal evolution of the solar fluxes and the depolarization ratios. As one could intuitively expect, the pollen vertical transport pattern, triggered by the turbulences caused by the heating/cooling of the ground, should follow with a given time delay (i.e. start after)





that of the solar flux. In order to quantify that time delay, $t$, for each day and for each of the two depolarization ratios, we have searched the value of $t$ that maximizes the correlation coefficient defined as:

$$r(t) = r\left(F(x), \delta(x-t)\right) \qquad (9)$$

where F is the solar flux and δ either $\overline{\delta^V}$ or $\overline{\delta^p}$. The optimized value of $t$ that maximizes the correlation

coefficient is called $t_{opt}$. The second and third rows of Figure 8 present the scatter plots of the solar flux vs. $\overline{\delta^V}$

and $\overline{\delta^p}$, respectively. We have represented the scatter plots without time delay (red colour, $r(t=0)$) and the

scatter plots with a time delay equal to $t_{opt}$ (blue colour, $r(t=t_{opt})$). In the first row of Figure 8 we have indicated

with an arrow the direction of translation of the $\overline{\delta^V}$ / $\overline{\delta^p}$ curves that $t_{opt}$ implies. The $\overline{\delta^V}$ r-values without

time delay are in the range $0.70 - 0.86$, which already indicates a good correlation between the solar flux and

$\overline{\delta^V}$. The r-values for $t=t_{opt}$ are significantly better as they range from 0.85 to 0.89. On 27 and 31 March $t_{opt} =$

-1 hour which indicates that the diurnal pattern of $\overline{\delta^V}$ follows that of the solar flux delayed approximately 1

hour. On 28 March $t_{opt} = +1$ hour which indicates that the diurnal pattern of $\overline{\delta^V}$ is ahead of that of the solar

flux approximately 1 hour. Let's recall that 28 March is one of the days with nocturnal pollen near-surface

activity and with the highest wind speeds. The maximum observed at 09 UT is due to a low layer of pollen (<

0.5 km) with relatively high values of $\delta^V$ (see Figure 3b). As far as $\overline{\delta^p}$ is concerned, the r-values without

time delay ($0.49 < \overline{\delta^p}\ r(t=0)-values < 0.89$) are also significantly improved when an optimized time

delay is applied ($0.77 < \overline{\delta^p}\ r(t=t_{opt})-values < 0.89$). The $t_{opt}$ values vary from 0 to +1 hour between those

of $\overline{\delta^V}$ and $\overline{\delta^p}$. After the optimized time delay is applied, $\overline{\delta^V}$ r-values are all greater or equal to the $\overline{\delta^p}$ r-

values. Differences vary between 0 and +0.08. These findings indicate that $\overline{\delta^V}$ is better correlated to the solar

flux than $\overline{\delta^p}$ is, which makes sense since both $\overline{\delta^V}$ and the solar fluxes depend on the molecules and the

particles, while $\overline{\delta^p}$ depends only on the particles.

### 6.   Conclusion

For the first time near-surface and columnar measurements of airborne pollen have been performed

continuously at a temporal resolution of one hour during a 5-day pollination event in a large European city. At

the peak of the event 2830 pollen and fungal spore grains were counted per cubic meter per day. *Platanus* and

*Pinus* pollen types represented together more than 80 % of the total concentration. Hourly concentration

maxima of 4700 and 1200 m$^{-3}$ h$^{-1}$ were found for *Platanus* and *Pinus*, respectively. Except on one day, the total

pollen concentration at ground level followed a clear diurnal cycle and was correlated positively with





temperature ($r = 0.95$) and wind speed ($r = 0.82$) but negatively with relative humidity ($r = -0.18$). These results indicate a strong dependence of pollen release upon the meteorological conditions, especially temperature and wind speed. As far as pollen AOD is concerned, its peaks were systematically associated with minima of relative humidity and maxima of temperature but they did not present a systematic dependence upon wind

speed.

The pollen AOD showed a clear diurnal cycle with maxima between 12 and 15 UT. The diurnal (9 – 17 UT) mean of $AOD_{pol}$ was 0.05 over the whole event and represented 29 % of the total AOD. However peaks of $AOD_{pol}$ and $AOD_{pol}$ / AOD of, respectively, 0.12 and 78 % were found on the hourly data. The diurnal mean volume and particle depolarization ratios in the pollen plume were 0.08 and 0.14, with hourly maxima of 0.18

and 0.33, respectively. The diurnal height of the pollen plume was found at 1.24 km on average with maxima varying in the range 1.47 – 1.78 km.

We have investigated the possible correlations between pollen near-surface concentration and columnar properties. Between concentration and AOD (be it total or pollen AOD) the correlation was rather poor which emphasizes that near-surface pollen release and columnar pollen dispersion are not timely correlated. $\overline{\delta^V}$ and

$\overline{\delta^p}$ were positively correlated with the total pollen concentration. The daily mean $\overline{\delta^V}$ and $\overline{\delta^p}$ $r$-values were, respectively, 0.34 and 0.41, with maxima of 0.81 reached on the first day of the event for both parameters. If we remove the days with nocturnal pollen near-surface activity, $\overline{\delta^V}$ and $\overline{\delta^p}$ $r$-values were greater than 0.68 and their difference smaller than 0.04. $\overline{\delta^V}$, and a fortiori $\overline{\delta^p}$, appeared to be an appropriate proxy for the total pollen concentration, especially when no pollen nocturnal activity is recorded.

The possible influence of solar radiations, which cause the atmospheric turbulences responsible for the aerosol vertical motion, on the vertical transport of pollen was examined by means of $\overline{\delta^V}$, $\overline{\delta^p}$ and solar fluxes measured during the three clear-sky days of the pollination event. Correlation coefficients better than 0.70 (0.49) were obtained for $\overline{\delta^V}$ ($\overline{\delta^p}$) vs. solar flux. In all cases we could find a time delay between the pattern of the pollen vertical transport and the one of the solar flux that could maximize the $r$-values. After the

optimized time delay was applied, correlation coefficients better than 0.85 (0.77) were obtained for $\overline{\delta^V}$ ($\overline{\delta^p}$) vs. solar flux. This study demonstrates that, in the absence of other depolarizing particles, the volume depolarization ratio is an excellent tool to track airborne pollen grains. On the one hand it is relatively well correlated with the pollen near-surface concentration which quantifies the pollen release; and on the other hand it is very well correlated with the solar fluxes on which the pollen vertical dispersion depends.

In our opinion this work puts forward two potential perspectives. First, relatively simple polarization-sensitive lidar systems (with at least two channels) able to produce continuously profiles of the volume depolarization ratio are very attractive tools for modellers to validate their pollen concentration forecasting models and/or perform data assimilation. The question was raised for $PM_{10}$ concentration by Wang et al. (2013). Second, the fact that large grains of pollen (of diameter ranging roughly in 20 – 70 µm) are capable of producing AOD of

0.12 raises the question of their effect in terms of radiative forcing. Otto et al. (2011) demonstrated that large



mineral dust particles with a diameter of 50 µm produced a radiative forcing at the surface almost four times greater than the one produced by particles with a diameter of 5 µm. Further research on that subject is definitely necessary.

**Acknowledgments**

Lidar data analysis were supported by the ACTRIS (Aerosols, Clouds, and Trace Gases Research Infrastructure Network) Research Infrastructure Project funded by the European Union's Horizon 2020 research and innovation programme under grant agreement n. 654169 and previously under grant agreement n. 262254 in the 7th Framework Programme (FP7/2007-2013); by the Spanish Ministry of Economy and Competitivity (projects TEC2012-34575 and TEC2015-63832-P) and of Science and Innovation (project UNPC10-4E-442)
and EFRD (European Fund for Regional Development); by the Department of Economy and Knowledge of the Catalan autonomous government (grant 2014 SGR 583). The authors also thank Xavier Lleberia and Marc Rico from the Agència de Salut Pública de l'Ajuntament de Barcelona for providing the PM$_{10}$ measurements, and the Facultad de Física of the Universitat de Barcelona for providing the meteorological measurements. Andrés Alastuey and Cristina Reche from the Instituto de Diagnóstico Ambiental y Estudios del Agua / Consejo
Superior de Investigaciones Científicas (IDÆA / CSIC) are thanked for their useful advice about the PM$_{10}$ measurements.

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



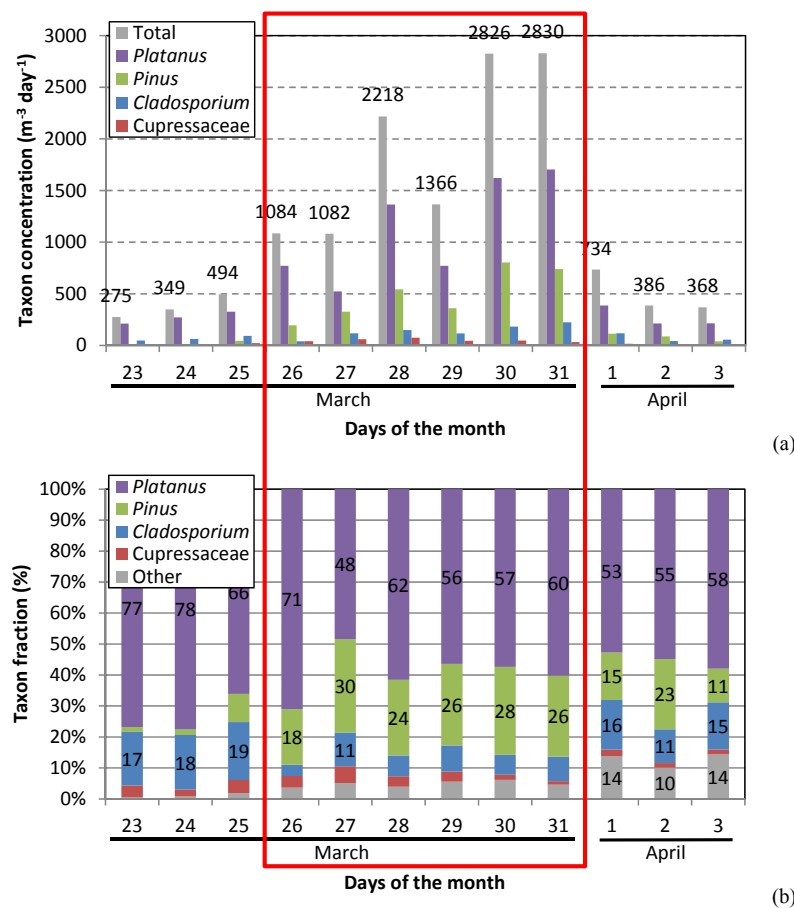

(a)

(b)

**Figure 1. (a)** Daily concentration of the four most abundant pollen (*Platanus*, *Pinus* and Cupressaceae) and fungal
5    spore (*Cladosporium*) taxa and total (pollen + spore), and **(b)** fraction of these four taxa during the period 23 March
– 3 April, 2015. The red rectangle indicates the intense pollination event. The values of the total concentrations are
reported in Figure 1a. The fractions higher than 10 % are reported in Figure 1b.





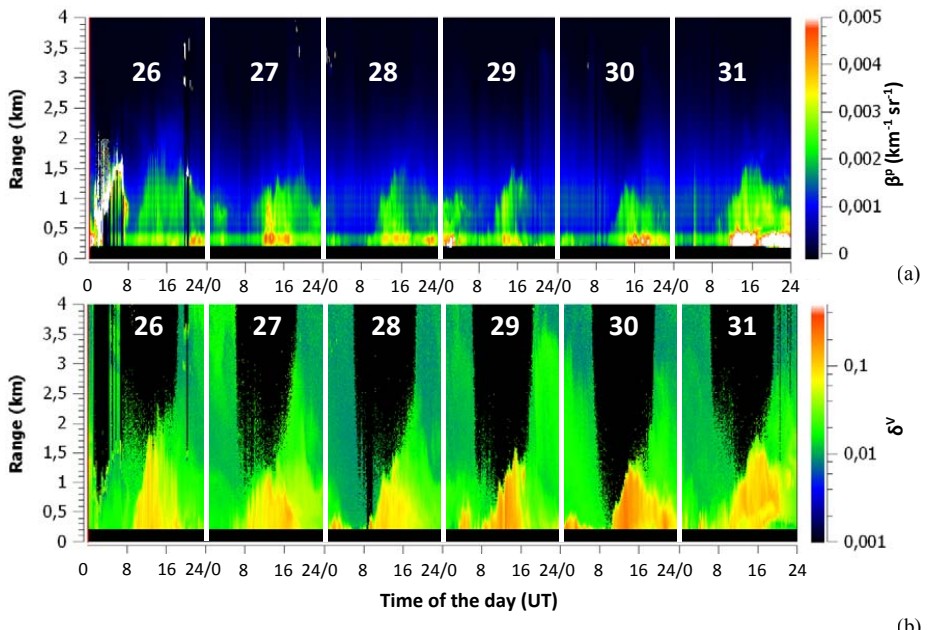

**Figure 2. 5-min. resolution time-range plots of the (a) particle backscatter coefficient ($\beta^p$) and (b) volume depolarization ratio ($\delta^v$) during 26 – 31 March, 2015.**





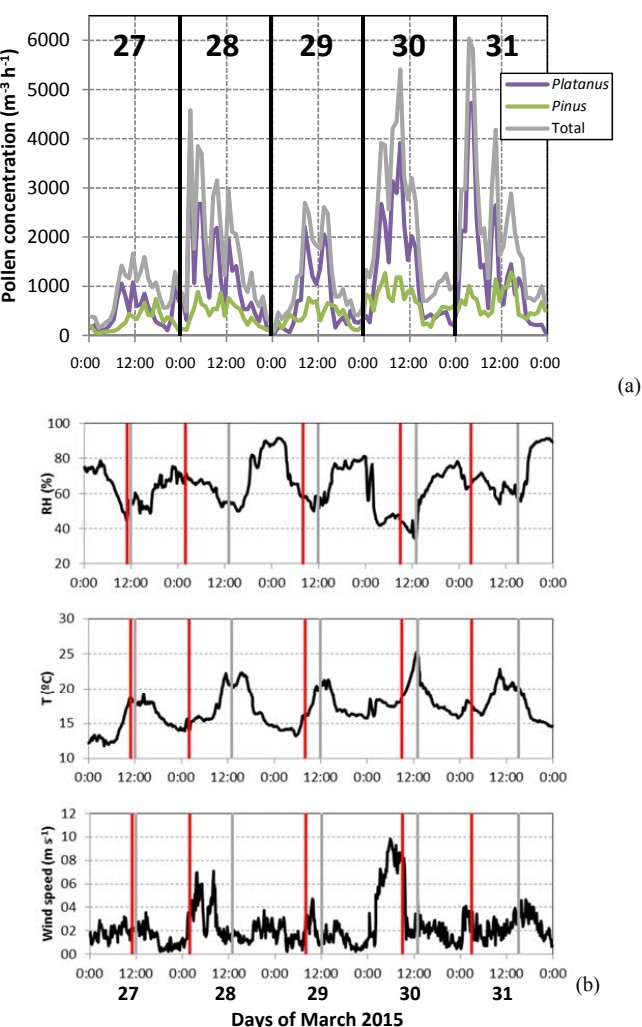

**Figure 3. Hourly temporal evolution (a) of the concentration of the two most abundant pollen taxa (*Platanus* and**

5      ***Pinus*) and (b) of the meteorological data: relative humidity (RH), temperature (T) and wind speed from 00 UT on**
**27 March until 24 UT on 31 March, 2015.  The red and grey vertical lines indicate the time of the maximum pollen**
**concentration and pollen optical depth, respectively, on each day.**



**Figure 4.** Diurnal time series of the hourly vertical distribution of the particle backscatter coefficient ($\beta^p$), the pollen backscatter coefficient ($\beta_{pol}$), the volume depolarization ratio ($\delta^V$) and the particle depolarization ratio ($\delta^p$) on (a) 27, (b) 28, (c) 29, (d) 30 and (e) 31 March, 2015.

The grey horizontal lines indicate the pollen layer height, $h_{pol}$.





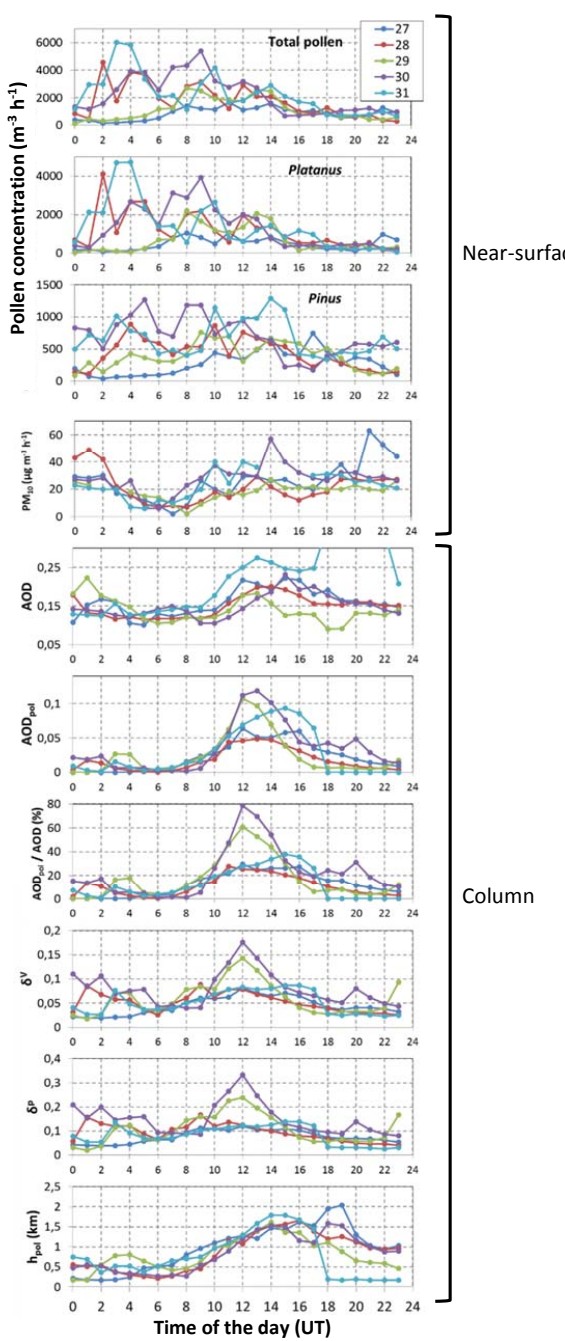

**Figure 5. Daily cycle for all five days of the pollination event of the total pollen, the *Platanus* and the *Pinus***

**concentration, $PM_{10}$, AOD, $AOD_{pol}$, $AOD_{pol}/AOD$, $\delta^V$, $\delta^P$ and $h_{pol}$ (see legend in the top plot).**





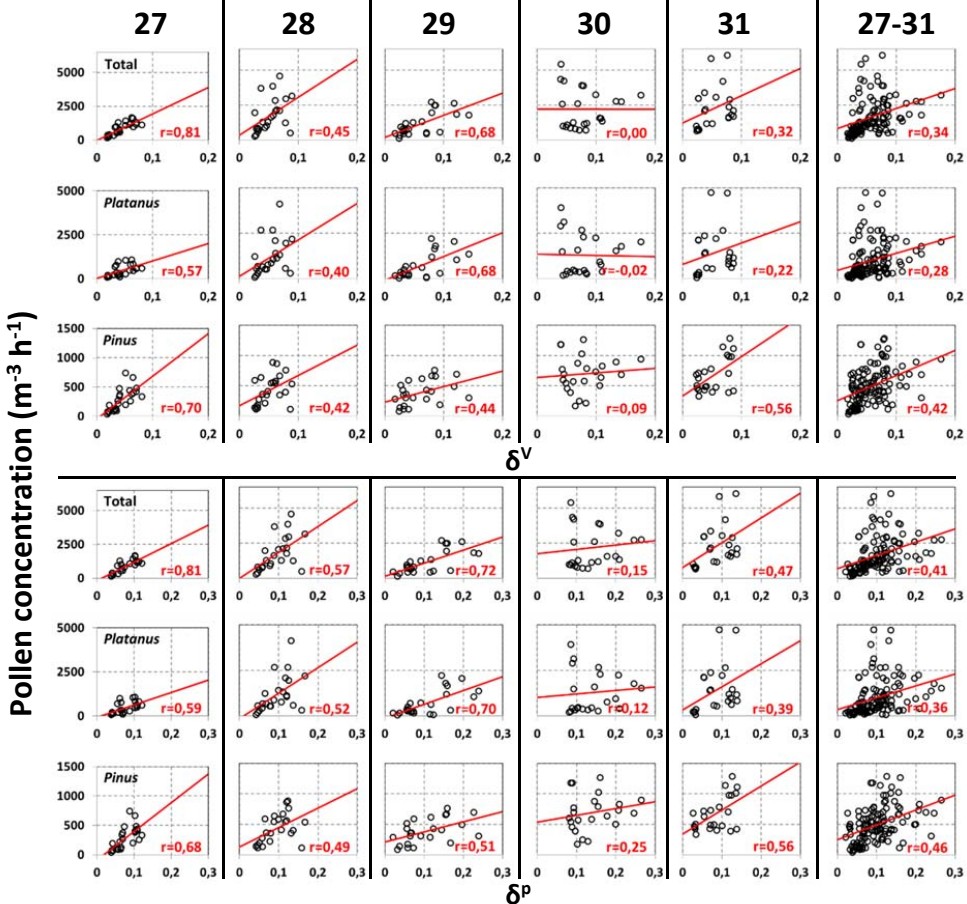

**Figure 6. Scatter plot of the pollen concentration (total, *Platanus* and *Pinus*) vs. the volume ($\overline{\delta^V}$) and the particle (**

$\overline{\delta^p}$ **) depolarization ratio integrated in the pollen plume for all five days of the pollination event. The linear**

**regression line is in red and the correlation coefficient, *r*, is reported.**





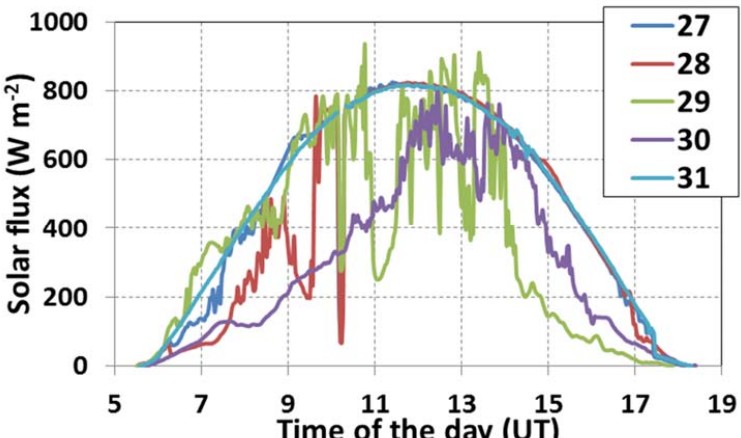

**Figure 7. Diurnal cycle for all five days of the pollination event of the total downward solar flux measured in the range 0.3 – 2.8 μm at ground level close to the lidar station.**





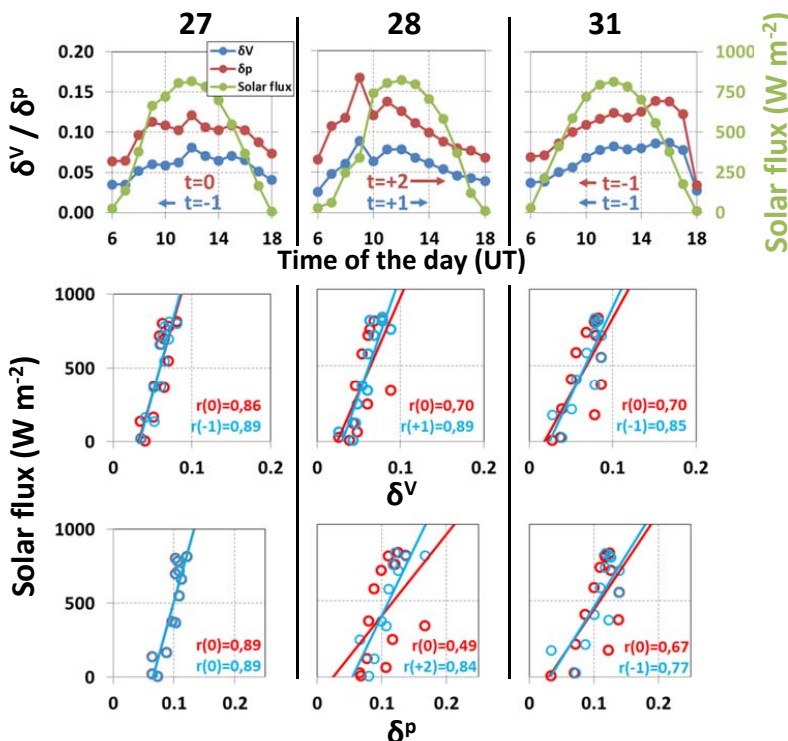

Figure 8. (top) Diurnal evolution of $\overline{\delta^V}$, $\overline{\delta^p}$ and solar fluxes on 27, 28 and 31 March between 06 and 18 UT; (center) scatter plot of the solar flux vs. $\overline{\delta^V}$; (bottom) scatter plot of the solar flux vs. $\overline{\delta^p}$. In each scatter plot, straight lines are linear regression lines; the red color corresponds to $t=0$; the blue color corresponds to $t=t_{opt}$. The values of $r(t)$ are reported in each scatter plot.



**0 – 24 UT**

|  | Units | 27 | 28 | 29 | 30 | 31 | 27-31 |
|---|---|---|---|---|---|---|---|
| Conc. total | m$^{-3}$ | 865 | 1770 | 1141 | 2201 | 2237 | 1643 |
| Conc. *Platanus* | m$^{-3}$ | 474 | 1181 | 690 | 1301 | 1384 | 1006 |
| Conc. *Pinus* | m$^{-3}$ | 286 | 433 | 387 | 700 | 668 | 495 |
| PM$_{10}$ | µg m$^{-3}$ | 25.7 | 21.4 | 18.2 | 27.5 | 22.8 | 23.2 |
| AOD |  | 0.16 | 0.15 | 0.14 | 0.15 | 0.24 | 0.17 |
| AOD$_{pol}$ |  | 0.02 | 0.02 | 0.02 | 0.04 | 0.03 | 0.03 |
| AOD$_{pol}$ / AOD | % | 12 | 11 | 17 | 23 | 12 | 15 |
| $\overline{\delta^V}$ |  | 0.04 | 0.05 | 0.06 | 0.08 | 0.05 | 0.06 |
| $\overline{\delta^p}$ |  | 0.08 | 0.10 | 0.11 | 0.15 | 0.08 | 0.10 |
| h$_{pol}$ | km | 0.98 | 0.87 | 0.81 | 0.85 | 0.77 | 0.86 |

**9 – 17 UT**

|  | Units | 27 | 28 | 29 | 30 | 31 | 27-31 |
|---|---|---|---|---|---|---|---|
| Conc. Total | m$^{-3}$ | 1251 | 1909 | 1779 | 2338 | 2364 | 1928 |
| Conc. *Platanus* | m$^{-3}$ | 640 | 1169 | 1133 | 1489 | 1321 | 1150 |
| Conc. *Pinus* | m$^{-3}$ | 458 | 547 | 580 | 635 | 832 | 610 |
| PM$_{10}$ | µg m$^{-3}$ | 23.9 | 17.6 | 18.6 | 34.8 | 31.7 | 25.3 |
| AOD |  | 0.19 | 0.17 | 0.14 | 0.16 | 0.23 | 0.18 |
| AOD$_{pol}$ |  | 0.05 | 0.03 | 0.05 | 0.06 | 0.07 | 0.05 |
| AOD$_{pol}$ / AOD | % | 23 | 20 | 34 | 40 | 27 | 29 |
| $\overline{\delta^V}$ |  | 0.07 | 0.06 | 0.09 | 0.10 | 0.08 | 0.08 |
| $\overline{\delta^p}$ |  | 0.11 | 0.11 | 0.15 | 0.18 | 0.12 | 0.14 |
| h$_{pol}$ | km | 1.31 | 1.22 | 1.18 | 1.11 | 1.37 | 1.24 |

**Table 1. Mean parameters during the pollination event calculated over the periods of 0 – 24 and 9 – 17 UT.**





**0 – 24 UT**

|  | 27 | 28 | 29 | 30 | 31 | 27-31 |
|---|---|---|---|---|---|---|
| $PM_{10}$ | 0.01 | -0.24 | -0.50 | -0.37 | -0.28 | -0.18 |
| AOD | 0.41 | -0.34 | -0.09 | -0.61 | -0.53 | -0.18 |
| $AOD_{pol}$ | 0.65 | 0.03 | 0.64 | -0.25 | 0.05 | 0.12 |
| $AOD_{pol}$ / AOD | 0.74 | 0.06 | 0.68 | -0.16 | 0.12 | 0.15 |
| $\overline{\delta^V}$ | **0.81** | 0.45 | 0.68 | 0.00 | 0.32 | 0.34 |
| $\overline{\delta^p}$ | **0.81** | **0.57** | **0.72** | **0.15** | **0.47** | **0.41** |
| $h_{pol}$ | 0.54 | -0.42 | 0.46 | -0.59 | 0.12 | -0.11 |

**9 – 17 UT**

|  | 27 | 28 | 29 | 30 | 31 | 27-31 |
|---|---|---|---|---|---|---|
| $PM_{10}$ | -0.22 | 0.19 | -0.28 | -0.32 | 0.22 | 0.18 |
| AOD | -0.23 | -0.24 | 0.41 | -0.88 | -0.58 | -0.13 |
| $AOD_{pol}$ | -0.19 | -0.05 | 0.55 | -0.26 | -0.50 | 0.12 |
| $AOD_{pol}$ / AOD | -0.09 | -0.03 | 0.58 | -0.03 | -0.46 | 0.18 |
| $\overline{\delta^V}$ | -0.05 | **0.77** | 0.65 | 0.07 | -0.63 | 0.33 |
| $\overline{\delta^p}$ | -0.08 | 0.72 | **0.68** | **0.23** | -0.56 | **0.37** |
| $h_{pol}$ | -0.22 | -0.68 | -0.05 | -0.69 | -0.41 | -0.35 |

**Table 2. Correlation coefficients of total pollen during the pollination event calculated over the periods of 0 – 24 and 9 – 17 UT. Bold numbers indicate for each day the parameter with the highest positive correlation coefficient. Numbers in red indicate for each parameter the day with the highest positive correlation coefficient.**





**0 – 24 UT**

|  | 27 | 28 | 29 | 30 | 31 | 27-31 |
|---|---|---|---|---|---|---|
| $PM_{10}$ | -0.03 | -0.10 | -0.49 | -0.33 | -0.37 | -0.19 |
| AOD | 0.14 | -0.34 | 0.00 | -0.55 | -0.53 | -0.22 |
| $AOD_{pol}$ | 0.36 | -0.03 | 0.63 | -0.23 | -0.07 | 0.05 |
| $AOD_{pol}$ / AOD | 0.44 | -0.01 | 0.65 | -0.15 | 0.01 | 0.09 |
| $\overline{\delta^V}$ | 0.57 | 0.40 | 0.68 | -0.02 | 0.22 | 0.28 |
| $\overline{\delta^p}$ | **0.59** | **0.52** | **0.70** | **0.12** | **0.39** | **0.36** |
| $h_{pol}$ | 0.22 | -0.44 | 0.36 | -0.55 | 0.02 | -0.16 |

**9 – 17 UT**

|  | 27 | 28 | 29 | 30 | 31 | 27-31 |
|---|---|---|---|---|---|---|
| $PM_{10}$ | -0.08 | 0.18 | -0.23 | -0.35 | -0.03 | 0.04 |
| AOD | -0.23 | -0.19 | 0.57 | -0.85 | -0.76 | -0.33 |
| $AOD_{pol}$ | -0.14 | -0.04 | 0.66 | -0.32 | -0.67 | -0.01 |
| $AOD_{pol}$ / AOD | -0.03 | -0.04 | 0.66 | -0.11 | -0.62 | 0.12 |
| $\overline{\delta^V}$ | 0.18 | **0.74** | **0.70** | -0.02 | -0.78 | 0.29 |
| $\overline{\delta^p}$ | **0.30** | 0.68 | **0.70** | **0.14** | -0.65 | **0.36** |
| $h_{pol}$ | -0.43 | -0.63 | 0.04 | -0.72 | -0.56 | -0.48 |

**Table 3. Idem as Table 2 for *Platanus*.**





**0 – 24 UT**

|  | 27 | 28 | 29 | 30 | 31 | 27-31 |
|---|---|---|---|---|---|---|
| $PM_{10}$ | 0.12 | -0.51 | -0.22 | -0.39 | 0.32 | -0.06 |
| AOD | 0.62 | -0.15 | -0.26 | -0.75 | -0.19 | 0.03 |
| $AOD_{pol}$ | 0.76 | 0.27 | 0.43 | -0.29 | 0.54 | 0.29 |
| $AOD_{pol}$ / AOD | **0.80** | 0.27 | 0.55 | -0.17 | 0.54 | 0.29 |
| $\overline{\delta^V}$ | 0.70 | 0.42 | 0.44 | 0.09 | **0.56** | 0.42 |
| $\overline{\delta^p}$ | 0.68 | **0.49** | 0.51 | **0.25** | 0.56 | **0.46** |
| $h_{pol}$ | 0.76 | -0.19 | **0.63** | -0.65 | 0.52 | 0.09 |

**9 – 17 UT**

|  | 27 | 28 | 29 | 30 | 31 | 27-31 |
|---|---|---|---|---|---|---|
| $PM_{10}$ | -0.16 | **0.49** | -0.21 | -0.14 | **0.81** | **0.26** |
| AOD | 0.13 | -0.04 | -0.61 | -0.86 | 0.27 | 0.02 |
| $AOD_{pol}$ | 0.00 | 0.14 | -0.43 | -0.04 | 0.32 | 0.15 |
| $AOD_{pol}$ / AOD | -0.03 | 0.17 | -0.29 | 0.21 | 0.31 | 0.14 |
| $\overline{\delta^V}$ | -0.45 | 0.45 | -0.18 | 0.30 | 0.20 | 0.19 |
| $\overline{\delta^p}$ | -0.79 | 0.38 | -0.07 | **0.44** | 0.10 | 0.22 |
| $h_{pol}$ | **0.59** | -0.42 | -0.33 | -0.54 | 0.29 | -0.08 |

**Table 4. Idem as Table 2 for *Pinus*.**