# Peer review of "Near-surface and columnar measurements with a Micro Pulse Lidar of atmospheric pollen in Barcelona, Spain"

_Atmospheric Chemistry and Physics, 2016_

## Referee Comment (RC1) · Anonymous Referee #1 · 15 Apr 2016

**Overall comment:**

The object of this paper is to analyze relationships between direct surface measurements of pollen and MPL data simultaneously observed, and also meteorological parameters including solar radiation for continuous pollination events in Barcelona.

The results are shown with many figures and tables, and discussed in detail on the phenomena. The results and discussion may be a little lengthy, e.g., as mentioned in the "P15, L20-L27". And the reviewer feels that the discussion is not satisfactory for readers because of less physical aspects of the relationships analyzed. For example, in section 5, they analyze and discuss the correlation with solar radiation. It is interesting that the correlation coefficients have been better when introducing the time-delay($t$). They show better results of correlation coefficient, when $t > 0$ and $t < 0$ for different days. What is the physical meaning of the time-delay ? Also why do these values show positive and negative in neighboring two days?

**Minor comments:**

P2, L22: Do authors have any reasons of two kinds of character types, normal and italic when written in the following manner, e.g.,

"*Ambrosia, Alnus, Artemisia, Betula, Corylus*, Chenopodiaceae, Cupressaceae/Taxaceae"

Such expression is also shown in other places. If these are common in this field, it is Okay, but if not, please change these into one type.

P6, L23: "In the second half of March 2015 a strong anticyclone positioned in the Atlantic Ocean west of the Portuguese coast generated southeasterly winds in the northeastern part of the Iberian Peninsula."

Is the wind direction correct? It must be northwesterly (not southeasterly) ?

P7, L8 ant other places: The unit for counting the number concentration of pollen even for the daily mean should be "$m^{-3}$", not "$m^{-3}\ day^{-1}$". This expression is physically wrong. So in this case they should express in the following manner, the daily mean concentration of pollen is xxx "$m^{-3}$". Also the unit of "$m^{-3}\ h^{-1}$" is wrong, shown in other sentences and figures. These should be changed.

P8, L28: Is "Figure 3b" correct? It looks like "Figure 5."

P9, L21-L24: "Logically a strong release of … are gathered." It is a little hard to understand it. Please modify the sentence into much easier expression.

P13, L11-L13: "… is not from local origin." Please show and explain some evidences/reasons of "not from local origin."

P14, L14: It is not so familiar with the following equation, "*Pinus* ($0.09 < \delta^V$ $r$ -values $< 0.70$ and $0.25 < \delta^V$ $r$ -values $< 0.68$),". Is it possible to change other expression for better understanding?

P15, L20-L27: The reviewer supposes that the content of detailed cloud conditions is not necessarily needed in this context, because the cloud type such as medium or high might be not directly related with pollination events.

P17, L35: The sentence of "Otto et al.(2011) … " may not be needed in this conclusion because the authors do not discuss on the radiative forcing in the main sections.

Figure 1: The unit of "$m^{-3}day^{-1}$" should be changed into "$m^{-3}$".

Figure 2: The notation of decimal point should be unified, such as "0.005" from 0,005". Also "$km^{-1}sr^{-1}$" should be unified into "$Mm^{-1}sr^{-1}$" because "$Mm^{-1}sr^{-1}$" is used in the text.

Figure 7: The colours of lines for $27^{th}$ and $31^{st}$ might be confused. These should be changed with other colours for discriminating clearly.

Figure 8: The colour "red" is used for both of $\delta^{P}$ and time-delay. In order to understand these figures easily, the colours of $\delta^{P}$ and $\delta^{V}$ should be modified from "red" and "blue" into others.

---

## Referee Comment (RC2) · Anonymous Referee #2 · 15 Apr 2016

Comment on "Near-surface and columnar measurements with a Micro Pulse Lidar of atmospheric pollen in Barcelona, Spain" by M. Sicard et al.

More and more attentions have been paid on biological aerosols due to their significant impacts on environment and climate. The article presents an investigation of near-surface and column characterization of atmospheric pollen in Barcelona, Spain, mainly by use of lidar measurements and sampling analysis. Moreover, impact of meteorological elements (e.g., RH, T and wind speed) and solar flux on atmospheric pollen load in the atmosphere was discussed in detail. The topic is of sufficient interest to the communities of study of atmospheric aerosol (especially bioaerosols), climate as well as human health. In general, I find this manuscript to be of interest for publication

and appropriate for this journal. There are several suggestions for improvement listed below that should be considered by the authors and the editors before publication.

1. In fact, there is uncertainty during measurements of atmospheric pollen and spores. As introduced by the authors, pollen and spores was identified using a fluorescent microscope. However, the results should be affected by other fluorescent particles. It will be easier for readers to understand if the authors briefly introduce how to identify pollen and spores, obtain their concentration as well as discuss its uncertainty.

2. In spring, dust aerosols could be long-range transported to Barcelona. And pollen, like dust aerosols, are coarse particles and shows strong backscatter signal and large depolarization ratio from lidar measurements. Even high mass concentration of PM10 also could be seen during dust events. So the authors should explain why this is a pollination event, not a dust event. How to distinguish dust particles from pollen and spores?

3. Page 4 line 24: incomplete sentence, "PâŤť and P|| represent the perpendicular and parallel backscatter powers respectively".

4. Page 4 line 30: please delete "linear". Strictly speaking, MPL depolarization ratio is not linear depolarization ratio.

5. Page 6 line 20: the authors should explain how to decide a threshold for estimating the vertical height of pollen plume. There will be better if the vertical height is estimated based on particle backscatter coefficient and depolarization ratio simultaneously.

6. A peak of particle backscatter coefficient is always found at near surface ($\sim$300m), but not for depolarization ratio profiles. Overlap correction is very important before retrieval of lidar observation data, especially within boundary layer. So probably the correction is not proper, then cause this problem. Please carefully check processing of lidar data.

7. Page 7 line 3: add "a day" to the end of "...pollen and fungal spore per cubic".

8. Page 8 line 15: use abbreviation at the first time, "relative humidity (RH) and temperature (T)".

9. Figure 3: why does the total pollen concentration peak precede the AOD peak of pollen? In general, high particle concentration and RH cause large AOD. However, on 31 March maximum pollen concentration and AOD were found at 3 UT and 15 UT, respectively. RH is very close but large differences between total pollen concentration ($\sim$ 2 times). Please explain the reason.

10. Section 4: It is very important to fix a depolarization ratio of pollen when estimate its contribution ratio and backscatter coefficient. The authors reference results reported by other researchers. However, depolarization ratios are also affected by ambient RH. So please consider the factor and discuss uncertainty of contribution ratio and backscatter coefficient caused by artificially decided depolarization ratio.

11. Page 11 line 21: The AOD is not reliable from lidar data, by integrating the profile of backscatter coefficient in the whole column and multiplying the assumed lidar ratio. Why do not use AOD from co-located AERONET sun-photometer?

12. Page 17 line 31: add "are" to "...lidar systems (with at least two channels) are able to produce continuously profiles...".

13. Figure 2: please use particle depolarization ratio rather than volume depolarization ratio.

14. Figure 4: Too many, hard to get the points. Please 1) remove all total backscatter coefficient and volume depolarization ratio, just keep pollen backscatter coefficient and particle depolarization ratio; 2) only plot 3-4 panels per day, 9 panels are too many.

15. Figure 5: Same problem as fig 4, please remove panels of total AOD and volume depolarization ratio, and re-arrange the figure side by side. 16. Figure 6: Remove the upper panels (pollen concentration vs. volume depolarization ratio).

17. Figure 8: Depolarization ratio is used in the figure, but why do not use pollen

concentration or backscatter coefficients?

18. A paper about the vertical distribution of Asian dust measured by three MPL Lidars over Northwest China (Huang Z. et al., 2010) was published in JGR. Please reference this paper to increase reader understanding of lidar data retrieval and MPL performance. Furthermore, studies of fluorescent spectrum of atmospheric aerosols from a lidar spectrometer system with high spectral resolution (Sugimoto N. et al., 2012, OE) provides a new tools for investigating vertical structure of biological particles, which will be very useful for readers to understand remote sensing of bioaerosols.

————————————————

---

## Author Comment (AC1) · 18 May 2016

acp-2016-212, "Near-surface and columnar measurements with a Micro Pulse Lidar of atmospheric pollen in Barcelona, Spain" by Sicard et al.

**General comments from the authors**: First of all, we want to thank the two referees of our paper for their time and revision of our work.  There is a small change in the paper that we would like to indicate to the referees. We noticed that the pollen on the last day of the event, on March 31$^{st}$, after 18UT is not removed from the atmosphere (as it was said in the initial submission). A look at Figure 2b evidences it: a strong volume depolarization ratio persists in the ABL after 18UT but not below 0.5 km where it comes down to values typical of the local, background aerosol in Barcelona.  This non-depolarizing plume (< 0.5 km) is at the same time associated with high values of the backscatter coefficient (Figure 2a). We checked that the detection of the pollen plume height was erroneously found at the first range bin shown (near 0.16 km), and not near the top of the ABL as it should be. We corrected our script and ran it again for the last hours of March 31$^{st}$. This re-run results in changes in $h_{pol}$ and thus on all integrated parameters (AOD$_{pol}$, AOD$_{pol}$ / AOD, $\overline{\delta^V}$ and $\overline{\delta^p}$ ) only for March 31$^{st}$ after 18UT. We updated the following figures: 5, 6 and 8 and all four tables, as well as the discussion related to March 31$^{st}$.  We insist that these little changes only affect the data from March 31$^{st}$ after 18 UT, and do not change anything to the conclusions of the paper.

**Answers to Referee#1's comments**

Overall comment:
The object of this paper is to analyze relationships between direct surface measurements of pollen and MPL data simultaneously observed, and also meteorological parameters including solar radiation for continuous pollination events in Barcelona.
The results are shown with many figures and tables, and discussed in detail on the phenomena. The results and discussion may be a little lengthy, e.g., as mentioned in the "P15, L20-L27". And the reviewer feels that the discussion is not satisfactory for readers because of less physical aspects of the relationships analyzed. For example, in section 5, they analyze and discuss the correlation with solar radiation. It is interesting that the correlation coefficients have been better when introducing the time-delay(t). They show better results of correlation coefficient, when t > 0 and t < 0 for different days. What is the physical meaning of the time-delay ? Also why do these values show positive and negative in neighboring two days?
**Authors' reply: The idea behind the correlation analysis between the quantity of pollen dispersed in the atmosphere (parametrized by δV or δp) and the solar radiation was to see if the solar radiation, which during daytime creates convective ascendant motion of air masses, is a major factor of the vertical motion of pollen during daytime.  This idea is also reinforced by the fact that during nighttime, i.e. without solar radiation, pollen is usually not detected in the PBL.  The objective of the correlation analysis is well introduced in the first paragraph of Section 5.**
**The convective ascendant motion of air masses produced by the solar radiation reaching the ground is not instantaneous; it occurs with a determined time delay.  The time delay introduced in the paper has the exact same meaning, and the question that we want to answer is: what is the time phase difference between the variations of the solar radiation and the ones of δV or δp?  We find that the time delay is between 0 and -1, which indicates a time phase difference between 0 and 60 minutes.  However on 28 march we have a surprising result: the δV or δp variations are ahead of time (t is positive) w.r.t. the solar radiation.  The probable explanations for it, and expressed in the text, are that 28 March is one of the days with nocturnal pollen near-surface activity and with the highest wind speeds.  On that particular day, the transport of pollen in the atmosphere was already occurring before (and without) the morning solar radiation.**

Minor comments:

P2, L22: Do authors have any reasons of two kinds of character types, normal and italic when written in the following manner, e.g., "Ambrosia, Alnus, Artemisia, Betula, Corylus, Chenopodiaceae, Cupressaceae/Taxaceae" Such expression is also shown in other places. If these are common in this field, it is Okay, but if not, please change these into one type.

**Authors' reply: This is related to botanical terms and the taxonomy nomenclature rules. The names of genus have to be written in italics or underlined, the names of families are written with the normal letter type. You can recognize the names of the families by the termination – aceae. It is necessary to maintain the types indicated in the submitted version in order to be correct from the botanical point of view of the paper.**

P6, L23: "In the second half of March 2015 a strong anticyclone positioned in the Atlantic Ocean west of the Portuguese coast generated southeasterly winds in the northeastern part of the Iberian Peninsula." Is the wind direction correct? It must be northwesterly (not southeasterly) ?

**Authors' reply: The referee is right. That was a mistake from our part. "southeasterly" has been replaced by "northwesterly" in the revised manuscript.**

P7, L8 ant other places: The unit for counting the number concentration of pollen even for the daily mean should be "m-3", not "m-3 day-1". This expression is physically wrong. So in this case they should express in the following manner, the daily mean concentration of pollen is xxx "m-3". Also the unit of "m-3 h-1" is wrong, shown in other sentences and figures. These should be changed.

**Authors' reply: Initially "day-1" and "h-1" were added in order to indicate clearly if we were talking of daily or hourly averages. However we agree with the referee that it makes the units of concentration not fully correct, since the concentration numbers are not assessed by dividing by a unit of time (day or hour). So all pollen concentrations have been expressed in m-3 in the revised manuscript.**

P8, L28: Is "Figure 3b" correct? It looks like "Figure 5."

**Authors' reply: Figure 3b is correct. In Fig. 3b, as mentioned in the caption, the red and grey vertical lines indicate the time of the maximum pollen concentration and pollen optical depth, respectively, on each day.**

P9, L21-L24: "Logically a strong release of … are gathered." It is a little hard to understand it. Please modify the sentence into much easier expression.

**Authors' reply: This sentence and the previous one were replaced by the following: "As expected, every day the $AOD_{pol}$ peak follows the total pollen concentration peak. Logically, in the case of pollen of local origin, not long-range transport, a peak of the amount of pollen in the atmosphere (parameterized by $AOD_{pol}$) can only happen if previously a strong release of pollen at the ground level (parametrized by the pollen concentration) has occurred.".**

P13, L11-L13: "… is not from local origin." Please show and explain some evidences/reasons of "not from local origin."

**Authors' reply: We have considerably modify the hypothesis related to the sudden removal of pollen on 31 March at 18h. The decrease of the pollen layer top height is associated with a strong increase of AOD, a strong increase of $\beta^p$ in the first 0.5 km, a decrease of $\overline{\delta^V}$ and $\overline{\delta^p}$, an increase of RH but with no significant variation of the near-surface $PM_{10}$ level. The characteristics of the layer below 0.5 km are thus not from sea salt (because we have low depolarization ratios), they are also not directly from the aerosols formed at ground level (PM**

levels are unchanged).  We conclude to a water uptake (hygroscopic growth) of the already lofted particles and put it as a hypothesis in the text:

"Finally the sharp decrease of $h_{pol}$ on 31 March at 18 UT is an indication of the sudden removal of the pollen from the ABL.  This decrease of $h_{pol}$ is associated with a strong increase of AOD due to high values of $\beta^p$ (> 5 Mm$^{-1}$ sr$^{-1}$) in the first 0.5 km (see ¡Error! No se encuentra el origen de la referencia.), a decrease of $\overline{\delta^V}$ and $\overline{\delta^p}$, an increase of RH (see ¡Error! No se encuentra el origen de la referencia.b) but with no significant variation of the near-surface PM$_{10}$ level.  All in all these results suggest that the removal of the pollen from the ABL on 31 March at 18 UT may have been accompanied by a possible hygroscopic growth of lofted particles, probably from local origin, below 0.5 km."

P14, L14: It is not so familiar with the following equation, "Pinus (0.09 <δV r -values < 0.70 and 0.25<δV r -values < 0.68),". Is it possible to change other expression for better understanding?
**Authors' reply: "R-values" are correlation coefficients (their definition is given the first time they appear in the manuscript in Section 4) and "XX R-values" are the correlation coefficients measured between the parameter XX and another parameter. We have used this notation in order to synthetize the discussion as "XX R-values" is much shorter than "the correlation coefficients of XX".  This notation also allows to treat "XX r-value" as a parameter and include it in equations like the one taken as example by the referee (0.09 < δ$_V$ r -values < 0.70).  In the absence of a standard notation to express in a more concise way the "correlation coefficient of XX" the authors have decided to leave the notation "r-values" as is the manuscript.**

P15, L20-L27: The reviewer supposes that the content of detailed cloud conditions is not necessarily needed in this context, because the cloud type such as medium or high might be not directly related with pollination events.
**Authors' reply: The most important for our analysis is that no low cloud was present during the pollination event (which could have made difficult the inversion of the lidar signals).  The referee is right that the details on the medium and high clouds for the two cloudy days (29 and 30 March) are not necessary and they have been deleted in the revised manuscript.  In turn the details have been kept for the three clear-sky days because we believe it is important to keep the detailed time of the presence of clouds (before 09:30UT on 27 March, before 10 and after 17UT on 28 March and after 17UT on 31 March) for the validation of the solar fluxes.**

P17, L35: The sentence of "Otto et al.(2011) … " may not be needed in this conclusion because the authors do not discuss on the radiative forcing in the main sections.
**Authors' reply: The effect of large pollen grains on the aerosol direct radiative forcing is only mentioned in the conclusions because it is a potential perspective that the authors would like to further investigate (in another paper).  A small sentence "Large pollen grains may behave the same." has been added at the end of the conclusions to get back to our topic: pollen (and not mineral dust).**

Figure 1: The unit of "m-3day-1" should be changed into "m-3".
**Authors' reply: All concentration units have been changed.**

Figure 2: The notation of decimal point should be unified, such as "0.005" from 0,005". Also "km-1sr-1" should be unified into "Mm-1sr-1" because "Mm-1sr-1" is used in the text.
**Authors' reply: The dot (.) decimal separator has been used in the whole manuscript.  The units of the backscatter coefficient have been unified as Mm-1 sr-1.**

Figure 7: The colours of lines for 27th and 31st might be confused. These should be changed with other colours for discriminating clearly.

**Authors' reply: The authors have a different opinion and thought that the colours were rather well chosen. To satisfy the referee, we have kept the very different colours dark blue, brown, green and violet and changed the light blue by black. For homogeneity, the same colour change was applied to Figure 7.**

Figure 8: The colour "red" is used for both of $\delta P$ and time-delay. In order to understand these figures easily, the colours of $\delta P$ and $\delta V$ should be modified from "red" and "blue" into others.

**Authors' reply: $\delta p$ is actually brown while the scatter plot with t=0 is effectively red. We have not applied the same line style as in Figure 4 (blue solid ($\delta p$) and dotted ($\delta V$) lines) because the symbols (solid circles) of Figure 8 would make difficult to differentiate them.**

---

## Author Comment (AC2) · 18 May 2016

acp-2016-212, "Near-surface and columnar measurements with a Micro Pulse Lidar of atmospheric pollen in Barcelona, Spain" by Sicard et al.

**General comments from the authors**: First of all, we want to thank the two referees of our paper for their time and revision of our work. There is a small change in the paper that we would like to indicate to the referees. We noticed that the pollen on the last day of the event, on March 31$^{st}$, after 18UT is not removed from the atmosphere (as it was said in the initial submission). A look at Figure 2b evidences it: a strong volume depolarization ratio persists in the ABL after 18UT but not below 0.5 km where it comes down to values typical of the local, background aerosol in Barcelona. This non-depolarizing plume (< 0.5 km) is at the same time associated with high values of the backscatter coefficient (Figure 2a). We checked that the detection of the pollen plume height was erroneously found at the first range bin shown (near 0.16 km), and not near the top of the ABL as it should be. We corrected our script and ran it again for the last hours of March 31$^{st}$. This re-run results in changes in $h_{pol}$ and thus on all integrated parameters (AOD$_{pol}$, AOD$_{pol}$ / AOD, $\overline{\delta^v}$ and $\overline{\delta^p}$ ) only for March 31$^{st}$ after 18UT. We updated the following figures: 5, 6 and 8 and all four tables, as well as the discussion related to March 31$^{st}$. We insist that these little changes only affect the data from March 31$^{st}$ after 18 UT, and do not change anything to the conclusions of the paper.

**Answers to Referee#2's comments**

Comment on "Near-surface and columnar measurements with a Micro Pulse Lidar of atmospheric pollen in Barcelona, Spain" by M. Sicard et al. More and more attentions have been paid on biological aerosols due to their significant impacts on environment and climate. The article presents an investigation of nearsurface and column characterization of atmospheric pollen in Barcelona, Spain, mainly by use of lidar measurements and sampling analysis. Moreover, impact of meteorological elements (e.g., RH, T and wind speed) and solar flux on atmospheric pollen load in the atmosphere was discussed in detail. The topic is of sufficient interest to the communities of study of atmospheric aerosol (especially bioaerosols), climate as well as human health. In general, I find this manuscript to be of interest for publication and appropriate for this journal. There are several suggestions for improvement listed below that should be considered by the authors and the editors before publication.

1. In fact, there is uncertainty during measurements of atmospheric pollen and spores. As introduced by the authors, pollen and spores was identified using a fluorescent microscope. However, the results should be affected by other fluorescent particles. It will be easier for readers to understand if the authors briefly introduce how to identify pollen and spores, obtain their concentration as well as discuss its uncertainty.
**Authors' reply: Pollen and spores counts were performed by specialist technicians using light microscope, not fluorescent microscope. The explanation of pollen and spores identification has been rewritten in order to better describe the methodology used (Lines 26-34 of the initial submission):**

**"The drum was changed weekly and the exposed tape was cut into seven pieces, each one corresponding to one day, which were mounted on separate glass slides. Pollen and spores were counted under light microscope, at 600X magnification. Daily average pollen and spore counts were obtained following the standardized Spanish method (Galán et al., 2007), consisting in to run four longitudinal sweeps along the 24 h slide for daily data, identifying and counting each pollen and spore type found. To obtain the hourly concentrations, twenty-four continuous transversal sweeps separated every 2mm along the daily-sample slide, were analyzed, since the drum rotates at a speed of 2 mm per hour. Daily and intra-diurnal (hourly)**

pollen and spore concentrations are obtained converting the pollen and spore counts into particles per cubic meter of air, taking into account the proportion of the sample surface analyzed and the air intake of the Hirst pollen trap (10 L min⁻¹)"

2. In spring, dust aerosols could be long-range transported to Barcelona. And pollen, like dust aerosols, are coarse particles and shows strong backscatter signal and large depolarization ratio from lidar measurements. Even high mass concentration of PM10 also could be seen during dust events. So the authors should explain why this is a pollination event, not a dust event. How to distinguish dust particles from pollen and spores?

**Authors'reply: The synoptic situations was not favourable at all for the transport of mineral dust to Barcelona. In the Supplement 1 at the end of this document we add the forecast of mineral dust load from the BSC-DREAM8b model as well as Hysplit backtrajectories. No dust transport is forecasted by the model and the backtrajectories, identical during the five days, clearly indicate a North Atlantic (and thus clean) origin of the air masses transported to Barcelona. As Supplement 1 is not provided in the revised manuscript, the following sentence has been added at the beginning of Section 3: "To confirm that mineral dust was not transported over Barcelona during the pollination event, we used the dust transport models BSC-DREAM8b v2 (Barcelona Supercomputing Center – Dust Regional Atmospheric Model 8 bins) and NMMB/BSC-DUST (Nonhydrostatic Multiscale Meteorological Model on the B grid / Barcelona Supercomputing Center – Dust), as well as HYSPLIT (Hybrid Single Particle Lagrangian Integrated Trajectory) backtrajectories (not shown).".**

3. Page 4 line 24: incomplete sentence, "Pâˇ Tt' and P|| represent the perpendicular and parallel backscatter powers respectively".

**Authors'reply: The sentence has been completed.**

4. Page 4 line 30: please delete "linear". Strictly speaking, MPL depolarization ratio is not linear depolarization ratio.

**Authors'reply: We agree with the referee that what Flynn et al. (2007) call MPL depolarization ratio, $\delta_{MPL}$, is not the linear depolarization ratio. However the depolarization product provided by our MPL system (MPL-4B-IDS Series) is the one given in our Eq. (3), which coincides with the linear depolarization ration taking into account that $\delta_{MPL} = P_{cr} / P_{co}$ and Eq. (1.6) of Flynn et al. (2007).**

5. Page 6 line 20: the authors should explain how to decide a threshold for estimating the vertical height of pollen plume. There will be better if the vertical height is estimated based on particle backscatter coefficient and depolarization ratio simultaneously.

**Authors'reply: Using $\beta_{pol}$ to extract the pollen top height is an estimation already based on the consideration of both the particle backscatter coefficient and the particle depolarization ratio (since both parameters are needed to retrieve $\beta_{pol}$). The value of the threshold was selected so that the integral of $\beta_{pol}$ (z) up to $h_{pol}$ represents at least 99 % of its integral over the whole column. It has been indicated with a new sentence at the end of Section 2.3: "This empirical threshold guarantees that the integral of $\beta_{pol}(z)$ up to $h_{pol}$ represents at least 99 % of its integral over the whole atmospheric column.".**

6. A peak of particle backscatter coefficient is always found at near surface (_300m), but not for depolarization ratio profiles. Overlap correction is very important before retrieval of lidar observation data, especially within boundary layer. So probably the correction is not proper, then cause this problem. Please carefully check processing of lidar data.

**Authors'reply: The referee is right that overlap correction is very important. In particular for the MPL system which has a full overlap at a distance > 3 km, it is indispensable to make such**

a correction if ones wants to study the low troposphere. All MPL profiles shown in the paper are obviously corrected from the overlap. The overlap function used is shown in the figure thereafter. As one can appreciate, it is a very smooth function which can not produce artefacts or artificial sudden changes in the corrected lidar signals, so that we do not believe that the variation below 0.5 km are due to the overlap correction.

We had a careful look at the overall MPL data pre-processing to verify if one of the pre-processing procedures could introduce artefacts in the lidar profiles. Besides checking the overlap correction, we looked at the background subtraction, and the deadtime and afterpulse corrections. The background signal is calculated with background "bins" acquired before the laser emission. These "bins" are clearly visible before the afterpulse peak on the raw returned powers, and are guaranteed to be free of any laser-induced atmospheric signal. The deadtime correction modifies very, very little the signal in the first few hundred meters. With the naked eye, this modification is undetectable. The profile used to correct from the afterpulse effect has values different from the background signal only for heights below 100 m, and for this reason, like the overlap correction, it could not explained the differences commented by the referee.

We believe that the increase of the backscatter coefficient below 0.5 km is actually real and reflects the increase of the aerosol concentration in the surface layer in Barcelona. The fact that the depolarization ratio is not varying as much as the backscatter coefficient is not surprising: the depolarization ratio is a semi-intensive parameter (while the backscatter coefficient is an extensive parameter) and if the main contributor to the depolarization, pollen, is well mixed and its contribution to the total aerosol load remains stable, it is to be expected that the depolarization will stay approximately constant with height. Now, the fact that the backscatter coefficient is varying while at the same time the depolarization ratio is not indicates that the increase of the backscatter coefficient is due to an equal relative increase of the concentration of both pollen and non-pollen particles.

[Figure]

7. Page 7 line 3: add "a day" to the end of ": : :pollen and fungal spore per cubic".
**Authors'reply: It has been corrected.**

8. Page 8 line 15: use abbreviation at the first time, "relative humidity (RH) and temperature (T)".
**Authors'reply: Abreviations RH and T have defined the first time they appear in the text.**

9. Figure 3: why does the total pollen concentration peak precede the AOD peak of pollen? In general, high particle concentration and RH cause large AOD. However, on 31 March maximum pollen concentration and AOD were found at 3 UT and 15 UT, respectively. RH is very close but large differences between total pollen concentration (_ 2 times). Please explain the reason.
**Authors'reply: Referee#1 had a similar question about this part of the manuscript. When the pollen is from local origin, not long range transport, the simple idea behind this statement is**

that the pollen has to be first released at ground level (concentration peak) before it disperses in the atmosphere (AOD peak). Our rationale is true only if pollen of local origin is considered, not long-range transport. This condition has been added in the text.

We have changed the two sentences starting by "As expected …" at the end of Section 3 by: "As expected, every day the AOD$_{pol}$ peak follows the total pollen concentration peak. Logically, in the case of pollen of local origin, not long-range transport, a peak of the amount of pollen in the atmosphere (parameterized by AOD$_{pol}$) can only happen if previously a strong release of pollen at the ground level (parametrized by the pollen concentration) has occurred".

10. Section 4: It is very important to fix a depolarization ratio of pollen when estimate its contribution ratio and backscatter coefficient. The authors reference results reported by other researchers. However, depolarization ratios are also affected by ambient RH. So please consider the factor and discuss uncertainty of contribution ratio and backscatter coefficient caused by artificially decided depolarization ratio.

**Authors'reply: We have assessed the impact of the uncertainty of $\delta_{pol}$ on $CR_{pol}$. A new paragraph has been added following the discussion on the selection of $\delta_{pol}$ in Section 4. The new paragraph is:**

**"To assess the impact of the uncertainty of the assumed pollen depolarization ratio, $\delta_{pol}$, on the uncertainty of the contribution ratio of the pollen to the total particle depolarization, $CR_{pol}$, we consider an error $\Delta\delta_{pol}$ in the pollen depolarization ratio and write:**

$$CR_{pol}(z) + \Delta CR_{pol}(z) = \frac{\delta^P(z) - \delta_{no-pol}}{1 + \delta^P(z)} \frac{1 + \delta_{pol} + \Delta\delta_{pol}}{\delta_{pol} + \Delta\delta_{pol} - \delta_{no-pol}}, \qquad \text{(1)}$$

**from which the relative error in $CR_{pol}$, $\varepsilon_{rCR_{pol}} = \dfrac{\Delta CR_{pol}}{CR_{pol}}$, can be found as:**

$$\varepsilon_{rCR_{pol}} = -\frac{\left(1 + \delta_{no-pol}\right)\Delta\delta_{pol}}{\left(1 + \delta_{pol}\right)\left(\delta_{pol} - \delta_{no-pol} + \Delta\delta_{pol}\right)}. \qquad \text{(2)}$$

**We have calculated $\varepsilon_{rCR_{pol}}$ for various values of the "true" pollen depolarization ratio ranging from 0.2 to 0.5, when $\delta_{no-pol} = 0.03$ and $\delta_{pol} = 0.4$ is assumed. In the range $-0.1 < \Delta\delta_{pol} < +0.1$ (i.e. 0.3 < "true" pollen depolarization ratio< 0.5) the contribution ratio error is limited to ± 20 % approximately."**

**For the referee we are attaching thereafter the figure (not shown in the paper) showing the error $\varepsilon_{rCR_{pol}}$ as a function of the "true" pollen depolarization ratio ranging from 0.2 to 0.5, when $\delta_{no-pol} = 0.03$ and $\delta_{pol} = 0.4$ is assumed. Datatips indicate that for a "true" pollen depolarization ratio ranging 0.3 – 0.5, -20 < $\varepsilon_{rCR_{pol}}$ < 20 % for an assumed $\delta_{pol} = 0.4$.**

[Figure]

11. Page 11 line 21: The AOD is not reliable from lidar data, by integrating the profile of backscatter coefficient in the whole column and multiplying the assumed lidar ratio. Why do not use AOD from co-located AERONET sun-photometer?

**Authors'reply: The referee is right: in general the AOD obtained from the integral of the backscatter coefficient profile multiplied by an assumed lidar ratio is not reliable. We can unfortunately not use co-located AERONET measurements to constrain the lidar inversion because the Barcelona AERONET sun-photometer was not in operation at the time of the pollination event. This is the reason why we had to choose a constant lidar ratio.**

**The selection of a proper lidar ratio (50 sr) was made given the time of year considered (March) and the type of aerosol observed (pollen). As explained at the end of Section 2.2, 50 sr falls in the range of the mean columnar lidar ratios, 46 – 69 sr, found in Barcelona during the period from February to April and calculated over a period of 3 years by Sicard et al. (2011). In that work the columnar lidar ratio was retrieved with the two-component elastic lidar inversion algorithm constrained with the aerosol optical depth from a sun-photometer (like the referee is suggesting to do in the present work). The second reason, and probably the most grounded one, is that Noh et al. (2013b) found a mean columnar lidar ratio of 50 ± 6 sr during a 6-day pollination event (mostly dominated by *Pinus* and *Quercus* pollen) in South Korea by using the two-component elastic lidar inversion algorithm constrained with the aerosol optical depth from a sun-photometer.**

12. Page 17 line 31: add "are" to ": : :lidar systems (with at least two channels) able to produce continuously profiles: : :".

**Authors'reply: The verb of this sentence is a little further. The full sentence is "First, relatively simple polarization-sensitive lidar systems (with at least two channels) able to produce continuously profiles of the volume depolarization ratio are very attractive tools for modellers to validate their pollen concentration forecasting models and/or perform data assimilation", and we believe it is correct as is.**

13. Figure 2: please use particle depolarization ratio rather than volume depolarization ratio.

**Authors'reply: To retrieve the particle depolarization ratio, one needs to invert the lidar signal to obtain the profile of either the backscatter or the extinction coefficient. With the Micro Pulse Lidar, which is a low energy system, such inversions typically require to average during periods of time of 1 hour. The visualization of the volume depolarization ratio with a 5-min. resolution, in opposition to 60-min. particle depolarization ratio, is an excellent way of showing qualitatively the pollen day-to-day evolution during the whole pollination event. The 60-min. profiles of the particle depolarization ratio are shown anyway in Figure 4. For these reasons, Figure 2 has been kept as a function of the volume depolarization ratio.**

14. Figure 4: Too many, hard to get the points. Please 1) remove all total backscatter coefficient and volume depolarization ratio, just keep pollen backscatter coefficient and particle depolarization ratio; 2) only plot 3-4 panels per day, 9 panels are too many.

**Authors'reply: To gain in readability Figure 4 has been re-formated as suggested by the referee. We have kept 4 profiles per day, every 3 hours at 9, 12, 15 and 18 UT. The profile of the particle backscatter coefficient has been removed. The profile of the volume depolarization ratio is a key parameter in the paper and for studying atmospheric pollen with a polarization-sensitive lidar system. Our study finally shows that it is almost as sensitive as the particle depolarization ratio to pollen with the advantage that its retrieval (Eq. 3 of the paper) is much more straightforward than that of the particle depolarization ratio (Eq. 4). For this reason we have wanted to keep the profile of the volume depolarization ratio in Fig. 4.**

15. Figure 5: Same problem as fig 4, please remove panels of total AOD and volume depolarization ratio, and re-arrange the figure side by side.

**Authors'reply: The panel of total AOD was removed from Figure 5. For the same reason given in the answer of the previous comment, the plot of the volume depolarization ratio was maintained in Figure 5.**

16. Figure 6: Remove the upper panels (pollen concentration vs. volume depolarization ratio).

**Authors'reply: For the same reasons given in the answer of the above two comments, we have wanted to maintain the plot of the volume depolarization ratio in Figure 6.**

17. Figure 8: Depolarization ratio is used in the figure, but why do not use pollen concentration or backscatter coefficients?

**Authors'reply: On request of the referee we have performed the correlation study between solar radiation and total pollen concentration and pollen AOD. In terms of correlation coefficient it is equivalent to use the pollen AOD or the integral of the pollen backscatter coefficient profile since both properties are related by a multiplicative factor. The following figure (not included in the paper) shows the results for the total pollen concentration. The correlation coefficients range between 0.31 and 0.73, and a large scatter of the points is observed. No time delay could be found to maximize the correlation coefficients. These results point out a poor correlation between solar radiation and near-surface pollen concentration.**

**The correlation study between solar radiation and pollen backscatter coefficient has been included in the revised manuscript: Figure 8 has been updated as well as the discussion at the end of Section 5.**

[Figure]

18. A paper about the vertical distribution of Asian dust measured by three MPL Lidars over Northwest China (Huang Z. et al., 2010) was published in JGR. Please reference this paper to increase reader understanding of lidar data retrieval and MPL performance. Furthermore, studies of fluorescent spectrum of atmospheric aerosols from a lidar spectrometer system with high spectral resolution (Sugimoto N. et al., 2012, OE) provides a new tools for investigating vertical structure of biological particles, which will be very useful for readers to understand remote sensing of bioaerosols.

**Authors'reply: Both references have been added in the revised manuscript. Thank you for indicating them to us!**

**Supplement 1**: BSC-DREAM8b mineral dust maps at 12UT and Hysplit backtrajectories arriving in Barcelona at 500, 1000 and 2000 m at 12UT